# Deploying Models to Non-participating Clients in Federated Learning without Fine-tuning: A Hypernetwork-based Approach

**Yuhao Zhou**[*], **Jindi Lv, Yuxin Tian, Dan Si, Qing Ye,**[†] **and Jiancheng Lv**[†]

School of Computer Science, Sichuan University

`sooptq@gmail.com`[*], `{yeqing,lvjiancheng}@scu.edu.cn`[†]

## Abstract

Federated Learning (FL) has emerged as a promising paradigm for privacy-preserving collaborative learning, yet data heterogeneity remains a critical challenge. While existing methods achieve progress in addressing data heterogeneity for participating clients, they fail to generalize to non-participating clients with in-domain distribution shifts and resource constraints. To mitigate this issue, we present HyperFedZero, a novel method that dynamically generates specialized models via a hypernetwork conditioned on distribution-aware embeddings. Our approach explicitly incorporates distribution-aware inductive biases into the model's forward pass, extracting robust distribution embeddings using a NoisyEmbed-enhanced extractor with a Balancing Penalty, effectively preventing feature collapse. The hypernetwork then leverages these embeddings to generate specialized models chunk-by-chunk for non-participating clients, ensuring adaptability to their unique data distributions. Extensive experiments on multiple datasets and models demonstrate HyperFedZero's remarkable performance, surpassing competing methods consistently with minimal computational, storage, and communication overhead. Moreover, ablation studies and visualizations further validate the necessity of each component, confirming meaningful adaptations and validating the effectiveness of HyperFedZero.

## 1 Introduction

Federated learning (FL) McMahan et al. (2017) enables privacy-preserving collaborative learning Li et al. (2020a) across decentralized clients' data Dean et al. (2012); Ben-Nun & Hoefler (2019); Shi et al. (2023); Zhou et al. (2024b). A key challenge of FL is addressing data heterogeneity among clients, arising from non-i.i.d. (*i.e.*, independent and identically distributed) characteristics, which can significantly impact model performance Ye et al. (2023); Zhang et al. (2021). Existing approaches primarily focus on client-side personalization, either by learning a personalized model Marfoq et al. (2021); Zhang et al. (2020) or by fine-tuning the global model (*e.g.*, basic fine-tuning McMahan et al. (2017), regularised fine-tuning Li et al. (2021); T Dinh et al. (2020); Shi et al. (2024), selective fine-tuning Arivazhagan et al. (2019); Collins et al. (2021), etc.) to better suit participating clients. These efforts have achieved remarkable progress in reducing impacts of data heterogeneity, leading to improved model performance for participating clients.

Nevertheless, this paradigm struggles to generalize when deploying trained models to previously unseen edge devices (*e.g.*, non-participating clients) with: (1) **in-domain distribution shifts** (*e.g.*, different class frequencies, feature shifts, etc.), and (2) **limited computational and communication resources** for fine-tuning. Additionally, as shown in Figure 1a, we observe that *state-of-the-art* methods in personalized FL perform exceptionally well on participating clients' local data but catastrophically fail when applied to non-participating clients with in-domain distribution shifts. This indicates that current methods lack zero-shot personalization capabilities for new data distributions even in the same domain, hindering the real-world applications of FL, especially for time-sensitive or resource-constrained services, like healthcare Nguyen et al. (2022b); Szatmari et al. (2020), recommendations Wahab et al. (2022), and edge computing Zhou et al. (2025a); Imteaj et al. (2021).

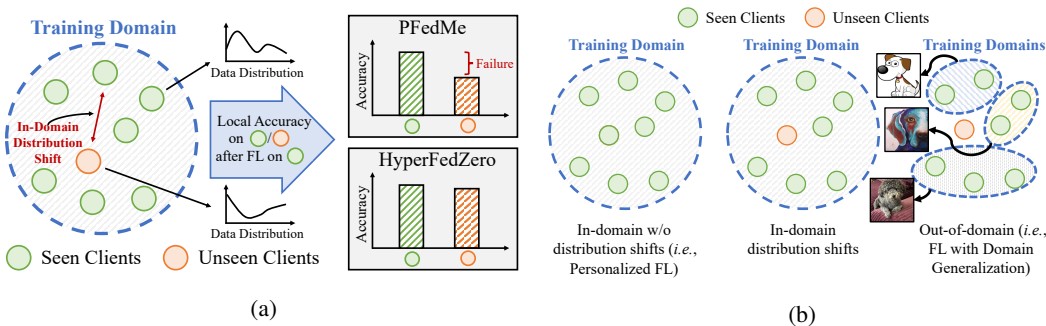

Figure 1: **[Left]** Previous *state-of-the-art* personalized FL methods perform well on seen clients but fail on unseen clients with in-domain distribution shifts (*e.g.*, different class frequencies, feature shifts, etc.). Conversely, HyperFedZero enables trained models to adapt to unseen clients by dynamically generating classifier parameters based on the input's distribution embeddings, overcoming in-domain distribution shifts without fine-tuning. **[Right]** Differences between in-domain without distribution shifts, in-domain distribution shifts and out-of-domain in FL.

To address the challenge, FedJets Dun et al. (2023) introduces Mixture-of-Experts (MoE Masoudnia & Ebrahimpour (2014)) architectures in FL, which turns the challenge of non-i.i.d. data into a blessing for expert specialization. Specifically, FedJets dynamically assigns different experts to different clients (whether seen or unseen) based on their unique data distributions, enabling zero-shot personalization on the fly. However, the server-side and client-side storage and computational requirements for managing extensive experts, as well as the need for frequent expert-parameter synchronization, create impractical bottlenecks.

Instead of following the previous approach of adapting each client's data separately via fine-tuning, we rethink the problem of deploying trained models to non-participating clients under strictly constrained resources from a novel perspective: *Can we directly encode distribution-aware inductive biases into the model's forward pass in FL without fine-tuning?* In this paper, we propose HyperFedZero, a hypernetwork-driven approach that dynamically generates the classifier parameters based on the input's distribution embeddings for improved zero-shot personalization under resource constraints. Specifically, rather than directly learning the mapping from inputs to labels, HyperFedZero learns the mapping from inputs to the optimal model parameters that can classify the inputs accurately. Additionally, the NoisyEmbed and the Balancing Penalty are also incorporated into HyperFedZero to further refine the extracted distribution embeddings by the distribution extractor to enhance robustness and prevent feature collapses Thrampoulidis et al. (2022).

Our contributions can be summarized as following:

1. We emphasize the inability to personalize models for unseen clients without fine-tuning leads to degraded performance when their data distributions, even within the same domain, differ from those observed during training (*i.e.*, In-domain distribution shifts). This limitation undermines the practicality of FL in dynamic environments with limited resources. To the best of our knowledge, this work could be one of the first attempts to mitigate this issue without incurring notable resource overheads.

2. We propose a novel hypernetwork-based approach, HyperFedZero, that directly encodes distribution-aware inductive biases into the model's forward pass. HyperFedZero begins by using a distribution extractor with NoisyEmbed and Balancing Penalty to capture robust and refined distribution embeddings from the input data. Then, a hypernetwork is conditioned on the extracted embeddings to dynamically generate classifier parameters. Finally, the input data are passed through classifiers to produce the final predicted labels.

3. Extensive experiments conducted across 7 datasets and 5 models demonstrate that HyperFedZero significantly outperforms competing methods in zero-shot personalization, while maintaining comparable model size and global and personalized performance. Additional ablation studies and visualizations further validate the superiority of HyperFedZero[1].

---

[1] https://github.com/Soptq/hyperfedzero-public

## 2 RELATED WORK

**Data heterogeneity in FL.** Data heterogeneity refers to differences in the statistical properties of data across clients, presenting a significant challenge in FL Ye et al. (2023); Zhang et al. (2021); Zhou et al. (2024a). Existing solutions fall into (i) *personalization*: FedPer Arivazhagan et al. (2019), FedProx Li et al. (2020b), PFedMe T Dinh et al. (2020), Per-FedAvg Fallah et al. (2020) learn client-specific models; (ii) *domain generalization*: COPA Wu & Gong (2021), FedDG Liu et al. (2021), FedSR Nguyen et al. (2022a), GA Zhang et al. (2023), FedIG Seunghan et al. (2024) train domain-invariant features for unseen domains;and (iii) *test-time adaptation*: FedTHE+ Jiang & Lin (2023), ATP Bao et al. (2023), TTA-FedDG Liang et al. (2025) optimize model predictions or parameters dynamically at test time to accommodate the distribution shift of target data. Neither stream handles *in-domain* distribution shifts under constrained resources common in practice.

**Hypernetworks.** A hypernetwork Ha et al. (2017); Chauhan et al. (2024); Wang et al. (2024) conditions on side information to emit target-network weights; recent chunked/diffusion variants cut its size. Recently, hypernetworks have gained considerable attention in the FL domain Shamsian et al. (2021); Chen et al. (2024); Shin et al. (2024); Yang et al. (2022). In FL it supports client personalization (pFedHN Shamsian et al. (2021)), communication compression (HyperFedNet Chen et al. (2024)), heterogeneous hardware (HypeMeFed Shin et al. (2024)) and device-specific CT models (HyperFed Yang et al. (2022)).

Recently, MoE-based FedJets Dun et al. (2023) tackled *in-domain* distribution shifts, but at the cost of significant computational and communication overhead. In contrast, OD-PFL Amosy et al. (2024) and PeFLL Scott et al. (2023) address this issue using hypernetwork to generate *client-level* weights. However, these methods introduce additional communication costs or privacy risks stemming from local data sharing. In comparison, our HyperFedZero generates *sample-level* weights locally (*i.e.*, entirely on client devices), enabling zero-shot adaptation for both seen and unseen clients without extra overhead or privacy concerns.

## 3 PROBLEM FORMULATION

Consider a FL training process with $N$ participating clients. Each client $i \in [0, N)$ owns a local dataset $D_i = (D_i^{\mathbf{x}}, D_i^{\mathbf{y}})$, and $(\mathbf{x}_i, \mathbf{y}_i) \sim D_i$ are drawn from the global instance space $\mathcal{X}$ and the global label space $\mathcal{Y}$, respectively. Additionally, each client $i$ maintains a classification model $c : \mathcal{X} \to \mathcal{Y}$ parameterized by global weights $\theta_c$ in the hypothesis space $\Theta_c$. The objective of FL is to find a $\theta_c$ that minimizes the overall losses across all participating clients, while maintaining data privacy, as shown by Equation 1.

$$\arg\min_{\theta_c} \sum_i^N w_i F_i((\mathbf{x}_i, \mathbf{y}_i), \theta_c), \tag{1}$$

where $F_i(\cdot)$ and $w_i$ are the local objective function and the aggregation weight of client $i$, respectively. The aggregation weight $w_i = |D_i| / \sum_k^N |D_k|$ helps combine clients' local losses into a global optimization target McMahan et al. (2017), where $|\cdot|$ is the size of the $\cdot$.

After obtaining $\theta_c$, the model is deployed to $M$ clients that did not participate in the FL process. Each client $j \in [0, M)$ has a local dataset $D_j$ which is drawn from $\mathcal{X}$ and $\mathcal{Y}$ (*i.e.*, shares the same domain as $D_i$) but exhibits different distributions (*e.g.*, different class frequencies, feature shifts, etc.). This results in in-domain distribution shifts, as the preferences of these non-participating clients were not considered during the training process in Equation 1. Therefore, a cold-start problem is introduced, as the model may not initially be well-suited to the data distribution of client $j$, leading to suboptimal performance. A simple workaround for this issue is to perform fine-tuning based on $\theta_c$. Nevertheless, it requires non-participating clients to have enough resources to handle additional local fine-tuning steps.

Intuitively, to avoid the aforementioned issues, we can directly condition the model's predictions on the distribution of the inputs. Specifically, this involves transforming Equation 1 to account for the distribution of $D_i$ during training, as illustrated by Equation 2.

$$\arg\min_{\theta_c} \sum_i^N w_i F_i((\mathbf{x}_i, \mathbf{y}_i), \theta_c, \mathbf{e}_i), \tag{2}$$

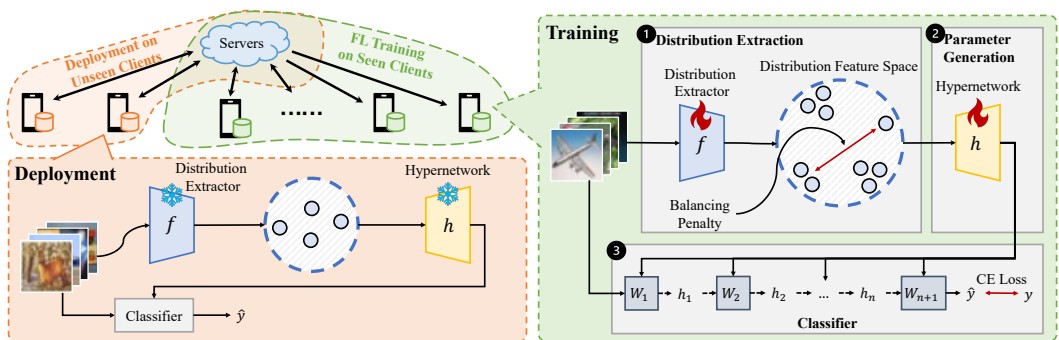

Figure 2: The general architecture of HyperFedZero consists of two main shared models: a distribution extractor $f$ and a hypernetwork $h$. During training, the distribution extractor $f$ first transforms the inputs into distribution embeddings, as shown in ❶. To prevent feature collapses, the NoisyEmbed and Balancing Penalty are applied. Then, in ❷, the hypernetwork $h$ generates chunked parameters based on the distribution embeddings. Finally, in ❸, a classifier $c$, initialized with generated parameters, is used to predict labels of the inputs. After training, frozen $f$ and $h$ can generate accurate classifiers that are well-suited for non-participating clients with in-domain distribution shifts.

where $\mathbf{e}_i$ is the distribution embeddings in the global distribution embedding space $\mathcal{E}$ extracted from $\mathbf{x}_i$. Nevertheless, how to properly obtain $\mathbf{e}_i$ and incorporate it into model predictions for non-participating clients with in-domain distribution shifts in FL remains an open problem. This is crucial for enabling effective zero-shot personalization.

## 4 OUR APPROACH

The general architecture of HyperFedZero is illustrated in Figure 2. In HyperFedZero, each client consists of a distribution extractor $f : \mathcal{X} \to \mathcal{E}$ parameterized by $\theta_f$ and a hypernetwork $h : \mathcal{E} \to \Theta_c$ parameterized by $\theta_h$. Specifically, for client $i$, the distribution extractor $f$ is responsible for generating inputs $\mathbf{x}_i$'s distribution embeddings $\mathbf{e}_i$ with a Balancing Penalty for preventing feature collapses. Meanwhile, based on $\mathbf{e}_i$, the hypernetwork $h$ generates dynamic $\theta_i^c$ for the classifier to predict the labels. In other words, instead of learning the mapping function directly from $\mathcal{X}$ to $\mathcal{Y}$, HyperFedZero lets clients first learn the mapping function from $\mathcal{X}$ to $\mathcal{E}$ to $\Theta_c$. Then, a classifier is initialized with generated $\theta_c \in \Theta_c$ to transform $\mathcal{X}$ to $\mathcal{Y}$.

### 4.1 DISTRIBUTION EMBEDDINGS EXTRACTION

For client $i$, the distribution extractor $f$ aims to embed the original inputs $\mathbf{x}_i$ into a normalized $P$-dimensional embeddings $\mathbf{e}_i \in \mathcal{E}$ that captures the geometric relationships (*i.e.*, similar embeddings imply similar distributions). Intuitively, similar to token embeddings in the NLP field Antoniak & Mimno (2018); Girdhar et al. (2023), where, with proper supervision from labels, the smoothness and continuity properties of neural networks naturally enable this embedding structure. However, we find a significant issue when simply obtaining $\mathbf{e}_i$ by $f(\mathbf{x}_i)$:

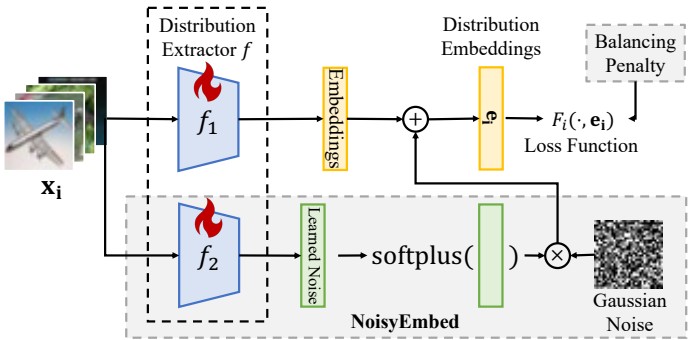

Figure 3: The NoisyEmbed and the Balacing Penalty are employed in the distribution extractor for improve distribution embeddings.

feature collapse. In this scenario, all $\mathbf{e}_i$ collapse into a narrow region within the embedding space.

This phenomenon arises because, during training, the local distributions of all clients can be sufficiently considered by Equation 1, as there are no non-participating clients at this time. In other words, all distributions are visible during training, minimizing the benefit of customizing models for invisible distributions. As a result, the distribution extractor tends to converge to a trivial solution, where all $\mathbf{x}_i$ are mapped to similar $\mathbf{e}_i$.

To mitigate the feature collapses issue, inspired by the load balance regulation in MoE Shazeer et al. (2017) where mechanisms to mitigate expert collapse and promote representation diversity have been analyzed and validated at scale Li et al. (2025); Mu & Lin (2025), we jointly employ NoisyEmbed and Balancing Penalty, as illustrated in Figure 3.

**NoisyEmbed** deliberately adds noises to $f(\mathbf{x}_i)$ for increased randomness and robustness, explicitly preventing feature collapses, as presented by Equation 3.

$$\mathbf{e} = \text{softmax}(f(\mathbf{x}_i; \theta_f) + z \cdot \text{softplus}(noisy(\mathbf{x}_i))), \tag{3}$$

where $z \in \mathcal{N}(0, 1)$. As it can be seen, NoisyEmbed employs an additional learnable global noisy network $f_2(\cdot)$ to customize the added noises to different inputs.

**Balancing Penalty** implicitly promotes exploration of the embedding space by incorporating Equation 4 into the loss function.

$$F_i(\cdot, \mathbf{e}_i) = F_i(\cdot) + \alpha \frac{var(\sum \mathbf{e}_i)}{mean(\sum \mathbf{e}_i)} + \beta \mathbf{E}(-\mathbf{e}_i \log \mathbf{e}_i), \tag{4}$$

where $\alpha$ and $\beta$ are two hyperparameters. In Equation 4, the first term encourages an even distribution of $\mathbf{e}_i$ across the embedding space Meanwhile, the second term fosters clustering along specific dimensions of the embedding.

## 4.2 Conditioned Prediction via Hypernetwork

Minimizing Equation 2 essentially maximizes the probability of correctly predicting the labels, *i.e.*,

$$\arg\max_{\theta_c} \sum_i^N w_i \text{Pr}(\mathbf{y}_i = \hat{\mathbf{y}}_\mathbf{i} | \mathbf{x}_\mathbf{i}; \theta_\mathbf{c}, \mathbf{e}_\mathbf{i}), \tag{5}$$

where $\hat{\mathbf{y}}_i$ represents the predicted label for client $i$ given $\mathbf{x}_i$, $\theta_c$ and $\mathbf{e}_i$. Thus, it is clear that we can approach the problem in two ways: either by conditioning the model's inputs on $\mathbf{e}$ or by conditioning the model's parameters on $\mathbf{e}$, *i.e.*,

$$\begin{cases} \arg\max_{\theta_c} \sum_i^N w_i \text{Pr}(\mathbf{y}_i = \hat{\mathbf{y}}_\mathbf{i} | \{\mathbf{x}_\mathbf{i}, \mathbf{e}_\mathbf{i}\}; \theta_\mathbf{c}), & \text{Opt. 1} \\ \arg\max_{\theta_c} \sum_i^N w_i \text{Pr}(\mathbf{y}_i = \hat{\mathbf{y}}_\mathbf{i} | \mathbf{x}_\mathbf{i}; \theta_\mathbf{c} | \mathbf{e}_\mathbf{i}), & \text{Opt. 2} \end{cases} \tag{6}$$

In HyperFedZero, we condition model's parameters on $\mathbf{e}$ (Opt. 2) for the following reasons, drawing on theoretical findings from Galanti & Wolf (2020): (1) In Opt. 1, a single classifier is responsible for making predictions on all inputs. This can be seen as making trade-offs along the Pareto front, limiting its flexibility and expressive power. (2) Additionally, in Opt. 1, the classifier may choose to ignore $\mathbf{e}_i$, which reduces the effectiveness of leveraging distribution embeddings and undermines modular, decoupled knowledge learning across different conditions. In contrast, Opt. 2 can be viewed as employing different models for different $\mathbf{e}_i$ in an explicit way, offering exponential parameter efficiency compared to Opt. 1. Sec. 6 further validates our design choices by empirically demonstrating that Opt. 2 consistently outperforms Opt. 1. However, Opt. 2 also introduces several challenges. First, Opt. 2 eliminates the knowledge sharing between classifiers as they are independent. Second, Opt. 2 requires managing multiple models on clients' devices, violating the principles of FL regarding model efficiency and resource usage. To alleviate these challenges, HyperFedZero employs a chunked hypernetwork $h$ to generate parameters incrementally. Specifically, the chunked hypernetwork $h$ is a simple MLP that has configurable architecture and chunk size. Given a predefined chunk size, $h$ splits the generated model's parameters into fixed-size groups (discarding the

remainder if any), and assigns a unique chunk embedding to each group. During parameter generation, $h$ generates each group chunk-by-chunk rather than all at once, with the MLP's output guided by the corresponding chunk embedding for each group. This enables the generation of different models based on $\mathbf{e}_i$ while maintaining shared global knowledge, as shown by Equation 7.

$$\arg\max_{\theta_c} \sum_{i}^{N} w_i \Pr(\mathbf{y}_i = \hat{\mathbf{y}}_\mathbf{i}|\mathbf{x}_\mathbf{i}; \mathbf{h}(\mathbf{e}_\mathbf{i}; \theta_\mathbf{h})). \tag{7}$$

In this way, HyperFedZero strikes a balance between flexibility and efficiency, allowing the system to leverage $\mathbf{e}$ and shared global knowledge while minimizing the overhead of managing multiple models on each client device.

## 4.3 ALGORITHM AND COMPLEXITY ANALYSIS

The pseudocode of HyperFedZero is presented in Algorithm 1 in the Appendix. In HyperFedZero, during each epoch, each client $i$ simultaneously minimizes the empirical risk on $D_i$ and the balancing penalty with distribution embeddings $\mathbf{e}_i$. This enables the extraction of meaningful embeddings, as well as distribution-aware parameters generation and prediction. Thus, no additional computational overhead is introduced, and the time complexity of HyperFedZero remains the same as FedAvg, equaling $\mathcal{O}(NEK)$. In terms of space complexity, the distribution extractor and the chunked hypernetwork can be very compact. This approach allows us to maintain a similar number of total parameters compared to directly using the classifier itself (*i.e.*, $|\theta_f| + |\theta_h| \approx |\theta_c|$). Therefore, HyperFedZero shares the same space complexity, $\mathcal{O}(N)$, with FedAvg as well.

## 4.4 PRIVACY IMPLICATION

HyperFedZero follows the standard FedAvg-style FL protocol: clients never share raw data, and the server aggregates client-side updates to form and broadcast a global model. In HyperFedZero, the only shared data are the parameters of the global hypernetwork and the distribution extractor, which is similar to sharing the global model in FedAvg. Consequently, the privacy attack surface is identical to FedAvg. Thus, privacy-preserving techniques developed for FedAvg, like secure aggregation Bonawitz et al. (2017) and differential privacy Dwork (2006), apply directly to HyperFedZero under the same protocol, without modifying the core method. Moreover, clients retain their raw data, distribution embeddings, and generated models produced by the hypernetwork entirely on-device. These personalized weights are never uploaded. Since the hypernetwork generates models only when conditioned on distribution embeddings, no third party can reconstruct that client's generated models without access to the client's data or a distributionally equivalent dataset.

## 5 EXPERIMENTS

**Datasets:** In line with community conventions Sattler et al. (2019); Zhou et al. (2023); Bernstein et al. (2018), our experiments utilizes five datasets: MNIST Deng (2012), FMNIST Xiao et al. (2017), EMNIST Cohen et al. (2017), SVHN Netzer et al. (2011), Cifar10 Krizhevsky et al. (2009), Cifar100 Krizhevsky et al. (2009) and Tiny-Imagenet Le & Yang (2015). To simulate the non-i.i.d. characteristic, each dataset is manually partitioned into multiple subsets using a Dirichlet distribution parameterized by $\alpha_d$, a method commonly employed in FL settings Wang et al. (2020); Li et al. (2022); Zhou et al. (2023). As a result, each client owns a distinct subset of the data, varying both in quantity and category.

**Models:** To cover both simple and complex learning tasks, five models are used in our experiments: Multi-Layer Perceptron (MLP), LeNet-S, LeNet, ZenkeNet Zenke et al. (2017), and ResNet He et al. (2016). Specifically, LeNet-S is a smaller version of LeNet, with reduced hidden layer dimensions. To enhance practicality, unlike previous work Sattler et al. (2019); Zhou et al. (2021; 2025b) that remove the batch normalization layers Ioffe & Szegedy (2015) and dropout layers Srivastava et al. (2014) in ResNet, we retain both of them without modification.

**Baselines:** In our experiments, we compare HyperFedZero against four categories of baselines: (1) Vanilla FL: Local, FedAvg McMahan et al. (2017); (2) In-domain without distribution shifts (*i.e.*, personalized FL): FedAvg-FT, FedProx Li et al. (2020b), Ditto Huang et al. (2021), pFedMe T Dinh

Table 1: The zACC, gACC and pACC comparisons (the higher the better) between settings. **Bold** marks the best-performing method in each comparison, underline marks the second best-performing method. HyperFedZero outperforms other baselines consistently.

| | MNIST | | | FMNIST | | | EMNIST | | | SVHN | | C-10 | C-100 | T-ImageNet |
| | MLP | LeNet-S | LeNet | MLP | LeNet-S | LeNet | MLP | LeNet-S | LeNet | ZekenNet | ResNet | | ResNet | |
|---|---|---|---|---|---|---|---|---|---|---|---|---|---|---|
| | | | | | | | | $N = 10$ | | | | | | |
| Local | 2.26 | 17.53 | 2.78 | 3.82 | 13.72 | 4.51 | 2.21 | 0.78 | 2.08 | 10.03 | 12.11 | 30.40 | 0.65 | 0.97 |
| FedAvg | 93.06 | 97.92 | 98.44 | 77.95 | 77.78 | 81.77 | 70.18 | 82.42 | 82.16 | 83.98 | 80.01 | 43.32 | 13.41 | 4.69 |
| FedAvg (g) | 93.83 | 97.72 | 98.40 | 85.48 | 86.11 | 87.69 | 71.05 | 82.09 | 83.31 | 85.64 | 83.37 | 44.27 | 14.41 | 6.89 |
| FedAvg (p) | 93.93 | 97.79 | 98.18 | 85.48 | 86.11 | 87.69 | 71.13 | 82.66 | 83.45 | 85.64 | 83.37 | 44.27 | 14.41 | 6.89 |
| FedAvg-FT | 89.24 | 92.01 | 90.28 | 57.99 | 48.44 | 71.35 | 47.27 | 28.52 | 57.81 | 46.68 | 35.61 | 32.39 | 3.52 | 1.34 |
| FedProx | 92.71 | 97.92 | 98.44 | 77.95 | 76.56 | 80.90 | 69.01 | 83.07 | 81.77 | 84.51 | 79.82 | 43.47 | 14.06 | 5.13 |
| Ditto | 92.53 | 98.09 | 98.26 | 77.08 | 77.08 | 80.03 | 68.62 | 82.29 | 80.73 | 82.36 | 68.42 | 35.80 | 8.98 | 4.54 |
| pFedMe | 93.23 | 97.92 | 98.26 | 77.78 | 77.08 | 78.82 | 69.40 | 81.64 | 81.77 | 82.62 | 75.20 | 38.78 | 11.46 | 4.39 |
| pFedHN | 26.91 | 17.36 | 10.94 | 26.56 | 13.37 | 18.40 | 9.25 | 1.17 | 2.47 | 6.32 | 6.58 | 30.54 | 4.69 | 0.89 |
| PerFedAvg | 93.23 | 97.92 | 98.26 | 78.30 | 77.26 | 80.90 | 70.05 | 82.68 | 81.90 | 45.25 | 78.52 | 43.32 | 13.28 | 5.73 |
| FedAMP | 89.41 | 91.67 | 90.80 | 59.55 | 51.04 | 71.35 | 47.53 | 30.86 | 58.33 | 47.01 | 35.42 | 32.67 | 4.17 | 1.12 |
| Scaffold | 94.27 | 98.26 | 98.61 | 78.47 | 78.30 | 80.73 | 71.61 | 82.94 | 82.94 | 84.83 | 81.48 | 47.30 | 15.63 | 8.26 |
| GA | 93.23 | 97.92 | 98.26 | 78.13 | 77.43 | 81.25 | 70.57 | 82.68 | 81.51 | 84.64 | 78.78 | 43.32 | 14.58 | 6.10 |
| FedSR | 94.79 | 97.92 | 98.44 | 79.69 | 81.94 | 81.94 | 74.09 | 82.94 | 83.07 | 85.42 | 79.49 | 43.18 | 11.59 | 6.25 |
| FedEnsemble | 84.38 | 92.53 | 92.36 | 65.10 | 64.58 | 65.45 | 11.46 | 58.07 | 70.57 | 59.31 | 77.38 | 51.14 | 11.98 | 6.17 |
| FedJETs | 93.75 | 96.88 | 98.26 | 77.43 | 78.47 | 81.77 | 69.14 | 73.70 | 83.33 | **87.04** | 77.47 | 54.69 | 13.15 | 4.98 |
| HyperFedZero | **95.49** | **98.09** | **98.78** | **82.99** | **83.68** | **82.29** | **76.82** | **83.20** | **83.59** | 85.09 | 82.36 | **57.24** | **16.06** | **9.08** |
| HyperFedZero (g) | 96.03 | 97.71 | 98.03 | 87.36 | 87.52 | 88.79 | 78.90 | 81.02 | 82.88 | 85.94 | 83.37 | 51.40 | 16.28 | 9.02 |
| HyperFedZero (p) | 95.93 | 97.82 | 98.21 | 88.08 | 88.14 | 89.24 | 78.13 | 81.53 | 82.46 | 85.00 | 83.03 | 51.00 | 18.31 | 9.44 |
| | | | | | | | | $N = 50$ | | | | | | |
| Local | 10.27 | 13.39 | 0.40 | 4.91 | 9.38 | 4.46 | 3.12 | 2.08 | 1.04 | 2.27 | 13.06 | 7.03 | 1.87 | 0.00 |
| FedAvg | 94.64 | 97.77 | 98.21 | 86.16 | 91.07 | 86.60 | 66.66 | 81.77 | 81.25 | 89.48 | 44.03 | 45.31 | 13.75 | 6.87 |
| FedAvg (g) | 93.60 | 97.89 | 98.15 | 85.42 | 86.04 | 87.27 | 70.67 | 81.65 | 83.68 | 87.17 | 49.61 | 42.85 | 16.60 | 6.25 |
| FedAvg (p) | 95.75 | 97.77 | 98.16 | 87.69 | 88.11 | 88.87 | 76.30 | 81.11 | 83.57 | 87.61 | 88.73 | 51.71 | 17.04 | 9.45 |
| FedAvg-FT | 87.95 | 83.93 | 93.30 | 84.37 | 67.85 | 71.42 | 45.83 | 28.64 | 63.02 | 48.58 | 41.47 | 29.68 | 5.00 | 0.31 |
| FedProx | 94.20 | 97.32 | 98.66 | 85.27 | 90.62 | 87.50 | 66.14 | 81.25 | 84.37 | 89.20 | 86.08 | 46.09 | 13.12 | 6.56 |
| Ditto | 94.20 | 96.88 | 98.21 | 84.82 | 91.07 | 87.50 | 65.62 | 79.16 | 81.25 | 84.37 | 69.31 | 33.59 | 3.75 | 0.31 |
| pFedMe | 94.20 | 96.43 | 98.66 | 84.82 | 87.50 | 86.60 | 61.45 | 74.47 | 83.33 | 80.68 | 81.53 | 31.25 | 6.25 | 2.81 |
| pFedHN | 92.41 | 63.33 | 7.58 | 70.08 | 47.77 | 18.30 | 44.79 | 7.81 | 5.20 | 77.27 | 44.03 | 21.09 | 1.25 | 0.93 |
| PerFedAvg | 94.20 | 97.77 | 98.66 | 85.26 | 90.18 | 86.16 | 67.18 | 81.71 | 83.85 | 90.05 | 88.07 | 35.93 | 16.25 | 5.62 |
| FedAMP | 89.29 | 89.29 | 93.30 | 84.37 | 77.67 | 72.76 | 50.00 | 41.66 | 64.06 | 50.28 | 42.33 | 23.43 | 4.37 | 0.62 |
| Scaffold | 94.64 | 98.21 | 98.55 | 87.94 | 87.50 | 87.50 | 70.83 | 81.25 | 84.89 | 90.34 | 88.64 | 45.31 | 16.87 | 10.31 |
| GA | 94.20 | 97.77 | 98.66 | 85.71 | 90.18 | 87.05 | 67.70 | 81.25 | 84.37 | 89.20 | 84.65 | 39.84 | 15.62 | 7.18 |
| FedSR | 95.98 | **99.11** | 97.32 | 87.94 | 87.50 | 88.39 | 70.31 | 80.72 | 83.85 | 90.62 | 80.39 | 39.84 | 10.62 | 5.31 |
| FedEnsemble | 82.14 | 94.64 | 92.86 | 74.55 | 72.32 | 75.00 | 13.54 | 59.37 | 64.58 | 65.91 | 85.79 | 50.00 | 15.00 | 6.25 |
| FedJETs | 95.98 | 97.77 | 98.21 | 87.05 | 83.93 | 90.17 | 74.49 | 78.12 | 83.33 | 81.25 | 81.25 | 53.13 | 18.75 | 5.31 |
| HyperFedZero | **97.32** | 98.66 | **99.55** | **91.52** | **91.51** | **92.86** | **77.60** | **83.33** | **87.00** | **91.47** | **92.04** | **61.79** | **19.37** | **14.68** |
| HyperFedZero (g) | 93.71 | 97.72 | 98.45 | 85.65 | 87.06 | 87.75 | 70.72 | 82.83 | 83.34 | 86.18 | 57.36 | 40.41 | 14.97 | 5.70 |
| HyperFedZero (p) | 96.08 | 97.83 | 98.21 | 87.92 | 87.77 | 89.07 | 76.40 | 82.12 | 84.12 | 87.56 | 87.06 | 52.40 | 17.36 | 12.56 |

et al. (2020), pFedHN Shamsian et al. (2021), PerFedAvg Fallah et al. (2020), FedAMP Huang et al. (2021); (3) In-domain with distribution shifts: FedEnsemble Shi et al. (2021), FedJets Dun et al. (2023); (4) Out-of-domain (*i.e.*, Federated Domain Generalization): Scaffold Karimireddy et al. (2020), GA Zhang et al. (2023), FedSR Nguyen et al. (2022a). Note that the Local baseline allows clients to perform local training without any communication, and FedAvg-FT enables clients to perform an additional one round of local fine-tuning after receiving the global model.

**Metrics:** For experiments involving $N$ participating clients, we first partition the dataset into $N + M$ non-i.i.d. subsets. Then, after training the global models on the $N$ participating clients, we report: (1) gACC: the top-1 accuracy evaluated on the global test set; (2) pACC: the averaged top-1 accuracy evaluated on the $N$ participating clients' local test set; (3) zACC: the averaged top-1 accuracy evaluated on the $M$ non-participating clients' whole set. Note that all three metrics are evaluated without any further fine-tuning after the training is completed.

**Implementation Details:** All experiments are conducted with $N = 10/50$ participating clients and $M = 5$ non-participating clients with a participation ratio of 1.0. The environment uses CUDA 11.4, Python 3.9.15, and PyTorch 1.13.0. The training involves $E = 500$ global epochs and $K = 5$ local iterations, with a global batch size of 800, learning rate $\eta = 0.001$, and $\alpha_d = 1.0$. In HyperFedZero, $\alpha = \beta = 1.0$, $P = 16$ by default. The size of hypernetworks (*i.e.*, the chunk size and the network

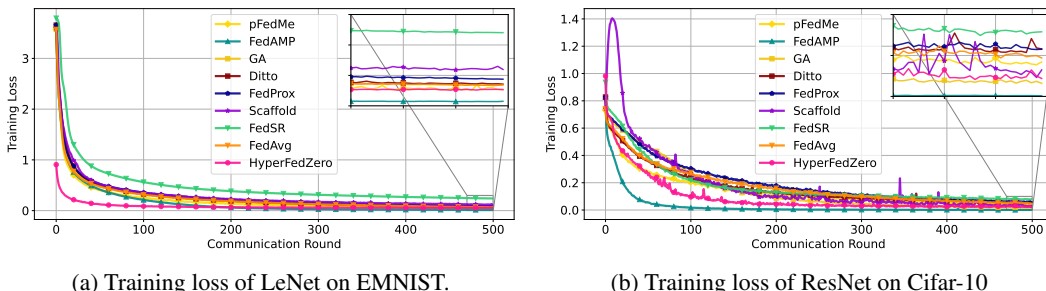

(a) Training loss of LeNet on EMNIST.
(b) Training loss of ResNet on Cifar-10

Figure 4: Training loss comparison between HyperFedZero with other methods. The convergence of HyperFedZero is comparable to others.

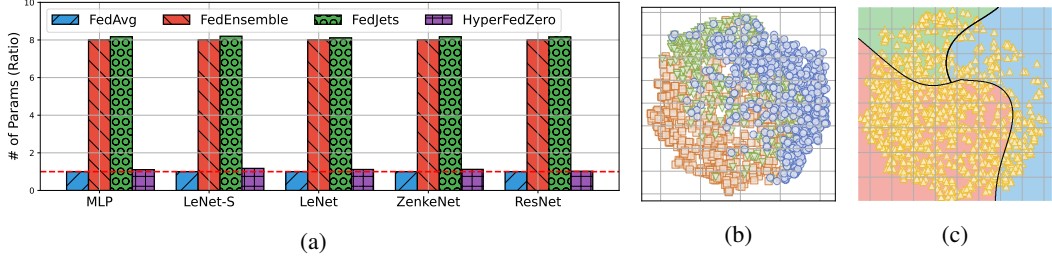

(a)
(b)
(c)

Figure 5: **(a)** Illustration of model sizes for FedAvg, HyperFedZero, and FedJets. HyperFedZero matches FedAvg in parameters and outperforms others in mitigating in-domain distribution shifts. **(b)** Visualized embeddings of three participating clients' data. Clearly, a decision boundary appears. **(c)** Visualized embeddings of a non-participating client's data. HyperFedZero directly generates specialized classifiers for different data, achieving optimal performance without local fine-tuning.

architecture) are tuned manually for each setting to ensure a similar number of total parameters compared to the classifier (*i.e.*, $|\theta_f| + |\theta_h| \approx |\theta_c|$). For other baselines, we adopt the hyperparameters as specified in their original papers.

## 6 ANALYSIS

Table 2: The zACC comparisons between Opt. 1 and Opt. 2 (Ours) in Equation 7, *i.e.*, two condition injection options. Opt. 2 improves flexibility and outperforms Opt. 1. **Bold** marks the best-performing results.

| | MNIST MLP | FMNIST MLP | EMNIST MLP | MNIST MLP | FMNIST MLP | EMNIST MLP |
|---|---|---|---|---|---|---|
| | $N = 10$ | | | | | |
| | $\alpha_d = 1.0$ | | | $\alpha_d = 0.1$ | | |
| FedAvg | 93.06 | 77.95 | 70.18 | 94.47 | 94.79 | 31.71 |
| Opt. 1 | 94.87 | 81.29 | 72.13 | 95.79 | 93.88 | 40.80 |
| Opt. 2 | **95.49** | **82.99** | **76.82** | **96.39** | **95.23** | **50.49** |
| | $N = 50$ | | | | | |
| | $\alpha_d = 1.0$ | | | $\alpha_d = 0.1$ | | |
| FedAvg | 94.64 | 86.16 | 66.66 | 89.58 | 82.63 | 62.50 |
| Opt. 1 | 95.08 | 84.82 | 74.37 | 90.83 | 78.75 | 65.36 |
| Opt. 2 | **97.32** | **91.52** | **77.60** | **92.36** | **85.41** | **68.05** |

**Main Results**: We compare the zACC of HyperFedZero with other baselines in Table 1. As can be seen, most personalized FL methods struggle to generalize to unseen data distributions within the same domain without additional fine-tuning. While federated domain generalization methods considerably enhance model generalization, they rely on training data from diverse and labeled domains, which does not apply to scenarios with in-domain distribution shifts. On the other hand, FedEnsemble and FedJETs significantly increase the number of trainable parameters and lack shared global information between sub-models, resulting in poor convergence. In comparison, HyperFedZero consistently achieves superior zACC across extensive settings with comparable gACC and pACC to others, indicating its ability to efficiently and effectively personalize the trained global model for unseen clients with in-domain distribution shifts, without any fine-tuning. Moreover, to compare the convergence performance of HyperFedZero with other methods, we visualize their respective training

Table 3: We conduct an ablation study on HyperFedZero's key hyperparameters to evaluate the effectiveness of our design choices. We report gACC, pACC, zACC, and $\Delta$ params (*i.e.*, the parameter difference between HyperFedZero and FedAvg) to provide a comprehensive analysis. Default settings are marked in gray . **bold** marks the best-performing results.

(a) The dimension of the $\mathbf{e}_i$. Large embedding dimensions lead to poor generalization.

(b) $\alpha$ in Equation 4. A moderate value of $\alpha$ yields the best performance.

(c) $\beta$ in Equation 4. A moderate value of $\beta$ yields the best performance.

| $P$ | gACC | pACC | zACC |
|---|---|---|---|
| \multicolumn{4}{c}{$N = 50; \alpha = 1.0$} | | | |
| 2 | 2.02 | 1.65 | 3.12 |
| 8 | 4.15 | 4.11 | 7.18 |
| 16 | **9.45** | **12.56** | **14.68** |
| 32 | 5.38 | 5.92 | 8.12 |
| 64 | 5.12 | 5.09 | 8.43 |
| \multicolumn{4}{c}{$N = 50; \alpha_d = 0.1$} | | | |
| 2 | 2.81 | 2.82 | 3.81 |
| 8 | 3.89 | 3.67 | 4.47 |
| 16 | **5.66** | **6.51** | **6.86** |
| 32 | 4.73 | 4.62 | 6.25 |
| 64 | 4.46 | 4.36 | 6.25 |

| $\alpha$ | gACC | pACC | zACC |
|---|---|---|---|
| \multicolumn{4}{c}{$N = 50; \alpha_d = 1.0$} | | | |
| 0 | 5.04 | 4.97 | 5.62 |
| 0.5 | 6.19 | 6.29 | 9.68 |
| 1 | **9.45** | **12.56** | **14.68** |
| 1.5 | 5.75 | 5.64 | 6.87 |
| 2 | 5.83 | 5.67 | 8.75 |
| \multicolumn{4}{c}{$N = 50; \alpha_d = 0.1$} | | | |
| 0 | 4.33 | 4.78 | 5.55 |
| 0.5 | **5.69** | 5.28 | 5.90 |
| 1 | 5.66 | **6.51** | **6.86** |
| 1.5 | 5.23 | 5.17 | 5.12 |
| 2 | 5.23 | 5.06 | 4.51 |

| $\beta$ | gACC | pACC | zACC |
|---|---|---|---|
| \multicolumn{4}{c}{$N = 50; \alpha_d = 1.0$} | | | |
| 0 | 5.73 | 5.71 | 8.43 |
| 0.5 | 5.96 | 5.69 | 8.43 |
| 1 | **9.45** | **12.56** | **14.68** |
| 1.5 | 6.45 | 8.2 | 10.12 |
| 2 | 6.47 | 6.29 | 10.31 |
| \multicolumn{4}{c}{$N = 50; \alpha_d = 0.1$} | | | |
| 0 | 5.41 | 5.15 | 4.16 |
| 0.5 | **5.67** | 5.54 | 4.51 |
| 1 | 5.66 | **6.51** | **6.86** |
| 1.5 | 5.56 | 5.39 | 5.16 |
| 2 | 5.38 | 5.54 | 5.55 |

(d) Hidden layer sizes in the hypernetwork $h$: Small $h$ limits model capacity, while large $h$ leads to poor convergence.

(e) The number of weights produced by the hypernetwork $h$ at a time ($\theta_c$ of the classifier is generated for multiple times)

| Archs of $h$ | gACC | pACC | zACC | $\Delta$ params |
|---|---|---|---|---|
| \multicolumn{5}{c}{$N = 50; \alpha = 1.0$} | | | | |
| $[100, 100]$ | 5.85 | 5.93 | 7.5 | -69.13% |
| $[300, 300]$ | **9.45** | **12.56** | **14.68** | +2.30% |
| $[500, 500]$ | 6.48 | 6.29 | 6.87 | +102.04% |
| \multicolumn{5}{c}{$N = 50; \alpha = 0.1$} | | | | |
| $[100, 100]$ | 5.20 | 4.95 | 4.86 | -69.13% |
| $[300, 300]$ | **5.66** | **6.51** | **6.86** | +2.30% |
| $[500, 500]$ | 5.11 | 4.97 | 6.25 | +102.04% |

| Chunk size | gACC | pACC | zACC | $\Delta$ params |
|---|---|---|---|---|
| \multicolumn{5}{c}{$N = 50; \alpha_d = 1.0$} | | | | |
| 144 | 5.74 | 5.77 | 7.50 | -27.01% |
| 288 | 7.19 | 6.93 | 9.37 | -22.79% |
| 576 | **9.45** | **12.56** | **14.68** | +2.30% |
| 1152 | 6.66 | 6.51 | 8.75 | +58.07% |
| 2304 | 5.11 | 5.28 | 7.81 | +182.09% |
| \multicolumn{5}{c}{$N = 50; \alpha_d = 0.1$} | | | | |
| 144 | 4.86 | 4.81 | 5.20 | -27.01% |
| 288 | 5.17 | 5.52 | 5.90 | -22.79% |
| 576 | 5.66 | **6.51** | **6.90** | +2.30% |
| 1152 | **5.92** | 5.61 | 5.90 | +58.07% |
| 2304 | 5.40 | 5.52 | 4.90 | +182.09% |

loss curves, as illustrated in Figure 4. As observed, HyperFedZero exhibits comparable convergence to the competing methods, empirically verifying the convergence guarantee of HyperFedZero.

Additionally, we present a visualization of the number of parameters stored in various model architectures for FedAvg, HyperFedZero, and FedJets in Fig 5a. As shown, HyperFedZero maintains a similar number of parameters compared to FedAvg, while delivering significantly superior performance in terms of zACC. This further verifies the effectiveness of HyperFedZero.

**Comparisons between Condition Options**: To assess the impact of conditioning the model's parameters on the embedding $\mathbf{e}$ (*i.e.*, Opt 2 in Equation. 7), we compare Opt 1 and Opt 2 in Table 2. From the table, we can observe that while Opt. 1 generally outperforms FedAvg, it underperforms in certain settings (*e.g.*, $N = 50$, $\alpha_d = 1.0$). This indicates that the injected conditioning does not generalize the global model effectively, and the added parameters may even degrade performance. In contrast, Opt. 2 consistently outperforms Opt. 1 and FedAvg across various values of $N$ and $\alpha_d$, highlighting its superior effectiveness.

**Embeddings Visualization**: We visualize the distribution embeddings using t-SNE Van der Maaten & Hinton (2008) after training with an MLP classifier on FMNIST in Figure 5c ($N = 50$, $M = 5$). The left panel shows the embeddings of data in three selected participating clients, while the right panel displays the embeddings of data in a non-participating client. As seen, a distinct decision

boundary is found in the left panel, indicating that HyperFedZero is capable of distinguishing data of different clients with distribution shifts. This demonstrates that HyperFedZero can dynamically generate specialized models based on embeddings when applied to non-participating clients, thereby enhancing performance. For instance, data in the green region of the right panel can be classified by generating a model similar to the one owned by the green client in the left panel.

**Ablation Study**: To investigate the impact of various hyperparameters on HyperFedZero's performance, we conduct ablation studies with a ResNet classifier on Tiny-ImageNet ($N = 50$), as shown in Table 3. These studies include ablations of $P$ (the dimension of $\mathbf{e}_i$, Table 3a), $\alpha$ and $\beta$ from Equation 4 (Table 3b and Table 3c), as well as the architectures and chunk size of the hypernetwork $h$ described in Section 4.2 (Table 3d and Table 3e).

In particular, the values of $P$, $\alpha$, and $\beta$ are critical in determining the model's ability to accurately capture and adapt to different data distributions, often requiring manual tuning through grid search. Empirically, the experimental results from both Table 3 and Table 1 suggest that $P = 16$, $\alpha = \beta = 1.0$ typically delivers "good" enough performance. Here "good" means by setting $P = 16$, $\alpha = \beta = 1.0$, HyperFedZero can be expected to achieve superior zero-shot personalization capability and maintain comparable global and personalized performance across diverse settings. Therefore, we believe $P = 16$, $\alpha = \beta = 1.0$ can be served as practical default values to begin with. On the other hand, the hyperparameters of $h$ influence the trade-off between model capacity and model size. Our empirical results show that tuning the hyperparameters of $h$ to maintain a similar number of parameters as FedAvg often yields the best performance.

## 7  CONCLUSION

In this work, we propose HyperFedZero, a novel FL method designed to address the critical challenge of generalizing trained global models to non-participating clients with in-domain distribution shifts. This is achieved by first learning discriminative distribution embeddings of different data with NoisyEmbed and Balancing Penalty. Then, these embeddings enable the chunked hypernetwork to dynamically generate personalized parameters without compromising privacy or requiring client-side fine-tuning. Empirical results across diverse settings also demonstrate HyperFedZero's superiority, outperforming other competing methods significantly while maintaining minimal computational and communication costs.

We believe this work bridges a critical gap in the practicality and scalability of FL by addressing the cold start problem during FL model deployment through zero-shot personalization. Like the open source culture, we believe this enables resource-constrained, non-participating clients to benefit from other clients' collaborative learning. In the future, we plan to extend HyperFedZero to incorporate diffusion-based parameter generation for even larger-scale real-world applications.

## ACKNOWLEDGMENTS

This work was supported by the National Natural Science Foundation of China under Grants 62427820 (National Major Scientific Instrument and Equipment Development Project) and 625B2122 (Basic Research Scheme for PhD Students), and in part by the Fundamental Research Funds for the Central Universities under Grant SCU2025D013.

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

## A MORE RELATED WORK

### A.1 DATA HETEROGENEITY IN FL

Data heterogeneity refers to differences in the statistical properties of data across clients, presenting a significant challenge in FL Ye et al. (2023); Zhang et al. (2021); Zhou et al. (2024a). To address this issue, previous research has mainly focused on two perspectives: adapting to in-domain data without distribution shifts (*i.e.*, personalized FL) and generalizing to out-of-domain data (*i.e.*, federated domain generalization). Specifically, personalized FL methods aim to learn a local model for each participant to accommodate its local data distribution. In particular, FedPer Arivazhagan et al. (2019) integrates a personalization layer into FL for customized fine-tuning. Conversely, FedProx Li et al. (2020b) introduces a proximal term that encourages the local models to be similar to the global model while also preserving the personalized updates. PFedMe T Dinh et al. (2020) further enhances personalized FL by incorporating Moreau Envelopes Moreau (1963), allowing the model to learn from global and local data distributions, and thereby improving generalization. Lastly, Per-FedAvg Fallah et al. (2020) utilizes a meta-learning strategy to develop an initialization for each client's local model that captures the structure of its local data. On the other hand, federated domain generalization approaches aim to improve model robustness across diverse and unseen domains by learning domain-invariant features. For instance, COPA Wu & Gong (2021) and FedDGLiu et al. (2021) apply multi-source domain generalization methods Nguyen et al. (2022a); Zhang et al. (2023) to FL by sharing classifiers and style distributions. Meanwhile, FedSR Nguyen et al. (2022a) proposes to learn a domain-invariant representation of the data with conditional mutual information and L2-norm regularizers. Later, GA Zhang et al. (2023) calibrates the aggregation weights in FL to achieve a tighter generalization bound. Recently, FedIG Seunghan et al. (2024) introduced client-agnostic learning for zero-shot adaptation, but it relies on multi-domain training data, which is often unavailable or unlabeled in real-world FL scenarios. A more recent line of work turns to test-time adaptation to achieve dynamical optimization of model predictions or parameters at test time to mitigate the distribution shift of target data. In particular, FedTHE+ Jiang & Lin (2023) proposes federated test-time head ensemble plus tuning to personalize FL models, enabling robustness against various test-time distribution shifts without test-time labeled data. ATP Bao et al. (2023) introduces the TTPFL setting and adaptively learns module-wise adaptation rates from source domain shifts to handle multiple distribution shifts in an unsupervised way. TTA-FedDG Liang et al. (2025) integrates test-time adaptation into FedDG, leveraging FedSPL with fast feature matching and knowledge distillation to boost generalization on unknown target clients.

Despite the promising performance, existing literature rarely explores in-domain distribution shifts in FL, as illustrated in Fig 1b. Namely, the data distribution shifts occur within the same domain, which is very common in real-world FL scenarios(*e.g.*, deploying an FL-trained package filtering model to a non-participating router Vamanan et al. (2010)). To address this issue, FedJets Dun et al. (2023) recently applies MoE to FL by dynamically assigning different experts to clients based on a learned gating function. However, it introduces additional resource overheads, limiting its practical application.

### A.2 HYPERNETWORK FOR PARAMETER GENERATION

The hypernetwork Ha et al. (2017) is a conditional meta neural network that generates all parameters for another network at once, enabling efficient model customization under varying conditions. However, generating all parameters simultaneously necessitates a sufficiently large hypernetwork, leading to significant resource overheads and unstable training. To address this, chunked hypernetwork Chauhan et al. (2024) and diffusion-based hypernetwork Wang et al. (2024) propose to incrementally generate parameters, substantially reducing the hypernetwork size without performance degradation. Moreover, hypernetworks can generalize well to unseen conditions Volk et al. (2022), facilitating diverse downstream applications like meta-learning Zhao et al. (2020); Beck et al. (2023); Cho et al. (2024, continual learning Von Oswald et al. (2019); Chandra et al. (2023); Hemati et al. (2023), and generative modeling Ratzlaff & Fuxin (2019); Schürholt et al. (2022); Do et al. (2020).

Recently, hypernetworks have gained considerable attention in the FL domain Shamsian et al. (2021); Chen et al. (2024); Shin et al. (2024); Yang et al. (2022). For instance, pFedHN Shamsian et al. (2021) trains a centralized hypernetwork on the server to dynamically generate personalized

models for clients based on their client embeddings. However, client embeddings only exist for participating clients, limiting pFedHN's adaptability to non-participating clients. Meanwhile, HyperFedNet Chen et al. (2024) reduces communication overhead in FL by compressing parameters of multiple models into a single hypernetwork. Additionally, HypeMeFed Shin et al. (2024) addresses hardware heterogeneity in FL by utilizing hypernetworks to generate different model architectures for different clients. Lastly, HyperFed Yang et al. (2022) employs hypernetworks to generate CT reconstruction models tailored to the specific parameters of CT machines. In comparison to these methods, HyperFedZero aims to generate parameters at a more granular level, customized for data samples rather than entire clients. This significantly enhances the model's adaptability for both participating and non-participating clients.

## B  ALGORITHM OF HYPERFEDZERO

---

**Algorithm 1** HyperFedZero

---

**Input**: global model parameters $\theta_f^t$ and $\theta_h^t$, local dataset $D_i = \{\mathbf{x}_i, \mathbf{y}_i\}$, learning rate $\eta_i$
**Parameter**: number of global epoch $E$, number of local iteration $K$, number of participating clients $N$
**Output**: global model parameters $\theta_f^E$ and $\theta_h^E$
**Clients:**
1:  **for** each client $i$ from 1 to $N$ **in parallel do**
2:      initialize $\theta_{i,f}^t = \theta_f^t$, $\theta_{i,h}^t = \theta_h^t$
3:      **for** each local iteration $k$ from 1 to $K$ **do**
4:          obtain $\mathbf{e}_i$ by Equation 3
5:          generate $\theta_c = h(\mathbf{e}_i; \theta_h^t)$
6:          compute loss $F_i(\cdot)$ by Equation 4
7:          $\theta_{i,f}^t = \theta_{i,f}^t - \eta_i \nabla_{\theta_{i,f}^t} F_i(\cdot)$
8:          $\theta_{i,h}^t = \theta_{i,h}^t - \eta_i \nabla_{\theta_{i,h}^t} F_i(\cdot)$
9:      **end for**
10:     **return** $\theta_{i,f}^t, \theta_{i,h}^t$
11: **end for**
**Servers:**
1:  initialize random $\theta_f^0, \theta_h^0$
2:  **for** each global epoch $e$ from 1 to $E$ **do**
3:      distribute $\theta_f^{e-1}, \theta_h^{e-1}$
4:      clients perform local training
5:      receive $\theta_{i,f}^{e-1}, \theta_{i,h}^{e-1}$
6:      $\theta_f^e = \sum_i^N \frac{|D_i|}{\sum_j^N |D_j|} \theta_{i,f}^{e-1}$
7:      $\theta_h^e = \sum_i^N \frac{|D_i|}{\sum_j^N |D_j|} \theta_{i,h}^{e-1}$
8:  **end for**
9:  **return** $\theta_f^E, \theta_h^E$

---

## C  NOTATIONS

The main notations in this paper are shown in Table 4.

## D  CONVERGENCE

Strictly speaking, the training phase of HyperFedZero is nothing more than a standard FedAvg applied to clients' local hypernetworks. As a result, the classical FedAvg convergence guarantees for smooth and potentially non-convex objectives Li et al. (2019); Haddadpour & Mahdavi (2019); Cho et al. (2020) carry over directly to our setting. Therefore, HyperFedZero inherits the same

Table 4: The glossary of notations

| Notation | Implication |
|---|---|
| $N$ | Total number of participated clients |
| $M$ | Total number of non-participated clients |
| $D_i$ | The local dataset of the $i$-th participated client |
| $\mathcal{X}$ | Global instance space |
| $\mathbf{x}_i \in \mathcal{X}$ | Instance from $D_i$ |
| $\mathcal{Y}$ | Global label space |
| $\mathbf{y}_i \in \mathcal{Y}$ | Labels from $D_i$ |
| $c : \mathcal{X} \to \mathcal{Y}$ | The classifier |
| $\Theta_c$ | Hypothesis space of the $c$'s parameters |
| $\theta_c \in \Theta_c$ | The $c$'s parameters |
| $f : \mathcal{X} \to \mathcal{E}$ | The distribution extractor |
| $\Theta_f$ | Hypothesis space of the $f$'s parameters |
| $\theta_f \in \Theta_f$ | The $f$'s parameters |
| $h : \mathcal{E} \to \Theta_c$ | The hypernetwork |
| $\Theta_h$ | Hypothesis space of the $h$'s parameters |
| $\theta_h \in \Theta_h$ | The $h$'s parameters |
| $\mathcal{E}$ | The global distribution embedding space |
| $\mathbf{e}_i \in \mathcal{E}$ | The distribution embeddings of the $i$-th client |
| $F_i(\cdot)$ | The local objective function of the $i$-th client |
| $w_i$ | The aggregation weight of the $i$-th client |

Table 5: The gACC comparisons (the higher the better) between settings ($\alpha_d = 1.0$). **Bold** marks the best-performing method in each comparison.

| | MNIST | | | FMNIST | | | EMNIST | | | SVHN | | C-10 | C-100 | T-ImageNet |
|---|---|---|---|---|---|---|---|---|---|---|---|---|---|---|
| | MLP | LeNet-S | LeNet | MLP | LeNet-S | LeNet | MLP | LeNet-S | LeNet | ZekenNet | ResNet | ResNet | ResNet | ResNet |
| *N = 10* | | | | | | | | | | | | | | |
| Local | - | - | - | - | - | - | - | - | - | - | - | - | - | - |
| FedAvg | 93.83 | 97.72 | 98.40 | 85.48 | 86.11 | 87.69 | 71.05 | 82.09 | 83.31 | 85.64 | 83.37 | 44.27 | 14.41 | 6.89 |
| FedAvg-FT | 88.84 | 91.20 | 91.58 | 73.18 | 60.24 | 80.70 | 52.62 | 37.34 | 63.27 | 51.11 | 35.69 | 34.06 | 4.03 | 1.38 |
| FedProx | 93.48 | 97.64 | 98.31 | 85.11 | 85.73 | 87.36 | 69.52 | 82.53 | 83.36 | 85.81 | 83.85 | 50.16 | 14.99 | 7.40 |
| Ditto | 93.28 | 97.66 | 98.11 | 85.11 | 85.16 | 87.22 | 69.49 | 82.08 | 82.44 | 83.74 | 71.95 | 40.97 | 11.28 | 3.72 |
| Scaffold | 94.65 | 97.85 | 98.40 | 86.09 | 84.91 | 87.70 | 73.43 | 83.53 | 84.03 | 85.82 | **84.17** | 50.68 | **16.91** | **9.78** |
| pFedMe | 93.74 | 97.50 | 98.13 | 85.38 | 85.51 | 87.16 | 69.73 | 81.75 | 82.83 | 83.31 | 79.78 | 45.61 | 11.99 | 6.26 |
| pFedHN | - | - | - | - | - | - | - | - | - | - | - | - | - | - |
| PerFedAvg | 93.81 | 97.69 | 98.36 | 85.50 | 85.68 | 87.61 | 70.96 | 82.69 | 83.32 | 50.50 | 83.09 | 49.35 | 13.46 | 6.63 |
| FedAMP | 88.72 | 91.03 | 91.95 | 73.38 | 61.64 | 80.76 | 52.78 | 37.85 | 63.36 | 46.42 | 36.14 | 34.92 | 4.28 | 1.36 |
| GA | 93.91 | 97.82 | 98.30 | 85.37 | 85.85 | 87.70 | 70.74 | **82.81** | 83.45 | 85.62 | 83.79 | 50.44 | 15.02 | 6.93 |
| FedSR | 95.15 | **97.92** | **98.69** | 86.16 | 87.42 | 88.38 | 74.67 | 81.96 | **84.61** | 86.13 | 82.31 | 46.16 | 12.48 | 8.30 |
| Ensemble | 81.73 | 92.02 | 94.10 | 70.92 | 74.77 | 76.19 | 19.02 | 60.38 | 68.87 | 60.03 | 79.19 | **54.22** | 15.87 | 8.88 |
| FedJETs | 94.12 | 96.28 | 98.22 | 84.54 | 84.50 | 87.64 | 70.96 | 75.12 | 83.90 | **86.70** | 79.61 | 47.97 | 14.14 | 6.62 |
| HyperFedZero | **96.03** | 97.71 | 98.03 | **87.36** | **87.52** | **88.79** | **78.90** | 81.02 | 82.88 | 85.94 | 83.37 | 51.40 | 16.28 | 9.02 |
| *N = 50* | | | | | | | | | | | | | | |
| Local | - | - | - | - | - | - | - | - | - | - | - | - | - | - |
| FedAvg | 93.60 | 97.89 | 98.15 | 85.42 | 86.04 | 87.27 | 70.67 | 81.65 | 83.68 | 87.17 | 49.61 | 42.85 | 16.60 | 6.25 |
| FedAvg-FT | 86.87 | 87.34 | 91.58 | 80.58 | 71.34 | 78.97 | 52.27 | 37.03 | 61.61 | 25.89 | 28.86 | 27.44 | 3.25 | 0.75 |
| FedProx | 93.05 | 97.74 | 98.08 | 85.15 | 85.42 | 86.95 | 69.48 | 81.27 | 83.37 | 87.04 | 86.82 | 43.77 | 16.38 | 6.18 |
| Ditto | 92.54 | 97.44 | 97.63 | 84.90 | 86.40 | 86.32 | 69.21 | 79.08 | 81.45 | 80.03 | 73.03 | 33.52 | 5.16 | 1.60 |
| Scaffold | 94.38 | 98.04 | **98.45** | 85.98 | 85.41 | 87.49 | 72.45 | **81.79** | **84.81** | **88.64** | **89.01** | 50.39 | **20.98** | **11.43** |
| pFedMe | 93.30 | 97.08 | 97.37 | 85.13 | 84.56 | 85.72 | 67.06 | 76.49 | 81.51 | 78.35 | 84.14 | 39.72 | 10.73 | 2.29 |
| pFedHN | - | - | - | - | - | - | - | - | - | - | - | - | - | - |
| PerFedAvg | 93.51 | **97.85** | 98.11 | 85.39 | 86.08 | 87.24 | 70.53 | 81.50 | 83.77 | 87.04 | 87.16 | 44.80 | 16.05 | 5.85 |
| FedAMP | 87.56 | 89.89 | 91.90 | 81.08 | 74.94 | 78.88 | 54.14 | 48.95 | 62.94 | 30.33 | 29.21 | 28.20 | 3.34 | 0.78 |
| GA | 93.18 | 97.82 | 98.12 | 85.28 | 85.83 | 87.09 | 70.52 | 81.49 | 83.77 | 87.20 | 87.39 | 42.88 | 15.85 | 6.03 |
| FedSR | 95.20 | 98.03 | 98.38 | 86.43 | 87.39 | 87.94 | 73.35 | 82.29 | 84.34 | 87.58 | 85.19 | 41.65 | 14.71 | 4.16 |
| Ensemble | 81.29 | 90.86 | 93.21 | 68.75 | 74.11 | 75.61 | 16.95 | 61.04 | 66.56 | 60.19 | 88.24 | **55.57** | 16.05 | 7.76 |
| FedJETs | 95.15 | 96.68 | 98.14 | 85.54 | 84.43 | 87.60 | 70.37 | 77.08 | 83.36 | 76.78 | 83.11 | 51.65 | 16.52 | 6.78 |
| HyperFedZero | **95.75** | 97.77 | 98.16 | **87.69** | **88.11** | **88.87** | **76.30** | 81.11 | 83.57 | 87.61 | 88.73 | 51.71 | 17.04 | 9.45 |

convergence rates as FedAvg, achieving linear convergence under strongly convex objectives and sub-linear rates in the non-convex case, even in the presence of aggregation noise.

Table 6: The pACC comparisons (the higher the better) between settings ($\alpha_d = 1.0$). **Bold** marks the best-performing method in each comparison.

| | MNIST | | | FMNIST | | | EMNIST | | | SVHN | | C-10 | C-100 | T-ImageNet |
|---|---|---|---|---|---|---|---|---|---|---|---|---|---|---|
| | MLP | LeNet-S | LeNet | MLP | LeNet-S | LeNet | MLP | LeNet-S | LeNet | ZekenNet | ResNet | | ResNet | |
| | | | | | | | $N = 10$ | | | | | | | |
| Local | 93.26 | 96.30 | 96.76 | 87.62 | 87.78 | 89.16 | 72.01 | 76.01 | 77.94 | 76.08 | 48.24 | 42.43 | 8.31 | 6.37 |
| FedAvg | 93.93 | 97.79 | 98.18 | 86.39 | 86.51 | 88.14 | 71.13 | 82.66 | 83.45 | 84.81 | 78.07 | 40.63 | 15.31 | 7.32 |
| FedAvg-FT | 93.26 | 96.27 | 96.78 | 87.81 | 87.81 | 89.15 | 71.98 | 75.88 | 78.17 | 76.16 | 49.02 | 47.91 | 13.34 | 6.37 |
| FedProx | 93.62 | 97.81 | 98.13 | 86.03 | 85.96 | 88.02 | 69.64 | 82.85 | 83.56 | 84.89 | 79.08 | 46.41 | 15.16 | 7.28 |
| Ditto | 93.41 | 97.68 | 98.05 | 86.05 | 85.74 | 87.87 | 69.83 | 82.06 | 82.30 | 83.24 | 66.02 | 37.21 | 11.06 | 3.80 |
| Scaffold | 94.76 | 98.25 | 98.30 | 86.26 | 86.26 | 88.19 | 73.54 | **83.42** | 84.00 | 85.15 | 82.98 | 49.59 | 18.23 | 9.74 |
| pFedMe | 93.88 | 97.70 | 97.96 | 86.27 | 86.01 | 88.04 | 70.26 | 82.07 | 82.68 | 83.26 | 74.59 | 41.53 | 12.59 | 6.51 |
| pFedHN | 93.13 | 94.00 | 95.76 | 86.06 | 82.20 | 86.76 | 65.35 | 51.18 | 73.70 | 69.67 | 63.90 | 42.59 | 11.47 | 5.95 |
| PerFedAvg | 93.92 | 97.75 | 98.14 | 86.48 | 86.31 | 88.10 | 71.03 | 82.64 | 83.27 | 76.07 | 79.19 | 45.75 | 13.50 | 6.90 |
| FedAMP | 93.22 | 96.41 | 96.75 | 87.71 | 87.61 | 88.95 | 71.69 | 76.09 | 78.36 | 72.75 | 48.36 | 47.35 | 13.45 | 6.12 |
| GA | 93.93 | 97.91 | 98.31 | 86.54 | 86.29 | 88.51 | 71.11 | 82.71 | 83.45 | 84.92 | 79.62 | 47.10 | 14.29 | 6.95 |
| FedSR | 95.87 | **97.99** | **98.61** | 86.37 | 87.44 | 89.24 | 74.38 | 81.74 | **85.48** | 85.38 | 77.39 | 41.16 | 12.81 | 8.61 |
| Ensemble | 82.96 | 92.19 | 94.04 | 71.43 | 75.34 | 77.46 | 19.22 | 61.52 | 69.03 | 58.51 | 78.25 | 49.59 | 15.22 | **9.78** |
| FedJETs | 93.93 | 96.17 | 98.15 | 85.01 | 84.26 | 88.69 | 71.86 | 75.51 | 83.72 | **85.49** | 76.39 | 45.71 | 14.64 | 6.81 |
| HyperFedZero | **95.93** | 97.82 | 98.21 | **88.08** | 88.14 | 89.24 | **78.13** | 81.53 | 82.46 | 85.00 | **83.03** | **51.00** | **18.31** | 9.44 |
| | | | | | | | $N = 50$ | | | | | | | |
| Local | 88.53 | 91.97 | 93.30 | 83.14 | 82.04 | 82.16 | 58.00 | 64.70 | 66.46 | 59.10 | 41.50 | 41.02 | 6.70 | 1.83 |
| FedAvg | 93.71 | 97.72 | 98.45 | 85.65 | 87.06 | 87.75 | 70.72 | 82.83 | 83.34 | 86.18 | 57.36 | 40.41 | 14.97 | 5.70 |
| FedAvg-FT | 88.53 | 91.97 | 93.28 | 83.14 | 82.11 | 82.24 | 58.00 | 64.84 | 66.51 | 59.04 | 40.98 | 40.85 | 6.12 | 2.08 |
| FedProx | 93.19 | 97.68 | 98.36 | 85.18 | 86.24 | 87.16 | 69.44 | 82.68 | 83.13 | 86.35 | 82.89 | 40.11 | 14.53 | 5.36 |
| Ditto | 92.79 | 97.20 | 97.83 | 85.21 | 87.57 | 86.52 | 68.78 | 80.28 | 80.93 | 79.49 | 66.28 | 31.25 | 4.57 | 1.57 |
| Scaffold | 94.68 | **98.07** | **98.71** | 86.10 | 86.24 | 88.00 | 72.82 | 82.99 | **84.83** | **87.78** | 85.09 | 46.68 | **19.30** | 11.11 |
| pFedMe | 93.43 | 97.17 | 97.75 | 85.28 | 85.87 | 85.95 | 67.11 | 76.81 | 80.96 | 77.83 | 78.58 | 36.87 | 10.16 | 2.24 |
| pFedHN | 92.68 | 75.69 | 92.34 | 82.34 | 71.26 | 79.93 | 58.81 | 16.98 | 55.19 | 70.98 | 57.36 | 35.11 | 4.71 | 2.62 |
| PerFedAvg | 93.58 | 97.72 | 98.46 | 85.55 | 87.20 | 87.68 | 70.21 | 82.73 | 83.38 | 86.30 | 83.01 | 40.94 | 14.77 | 5.45 |
| FedAMP | 88.56 | 91.98 | 93.29 | 83.13 | 82.20 | 82.29 | 58.23 | 64.95 | 66.45 | 59.51 | 40.56 | 40.50 | 6.68 | 1.98 |
| GA | 93.30 | 97.66 | 98.41 | 85.53 | 86.82 | 87.53 | 70.32 | 82.94 | 83.57 | 86.37 | 83.03 | 40.52 | 15.12 | 5.73 |
| FedSR | 95.39 | 97.72 | 98.49 | 86.61 | 87.39 | 88.90 | 72.80 | **83.22** | 84.69 | 86.33 | 81.26 | 37.71 | 13.62 | 4.10 |
| Ensemble | 80.76 | 91.07 | 93.74 | 69.76 | 75.15 | 76.17 | 18.10 | 60.73 | 65.98 | 58.99 | 82.78 | 49.90 | 14.51 | 8.48 |
| FedJETs | 95.19 | 96.86 | 98.22 | 85.47 | 84.21 | 87.59 | 69.61 | 78.23 | 83.12 | 76.59 | 79.45 | 50.14 | 16.09 | 6.19 |
| HyperFedZero | **96.08** | 97.83 | 98.21 | **87.92** | **87.77** | **89.07** | **76.40** | 82.12 | 84.12 | 87.56 | **87.06** | **52.40** | 17.36 | **12.56** |

# E    ADDITIONAL EVALUATION RESULTS

In this section, we present additional results for the proposed HyperFedZero and the baseline methods.

Specifically, Table 5 and Table 6 illustrate the gACC and pACC comparisons between HyperFedZero and other baseline methods. As shown, HyperFedZero achieves comparable performance to previous *state-of-the-art* approaches, while also exhibiting superior performance in zACC (as shown in the main paper), further reinforcing its overall superiority.

Additionally, we assess the performance of HyperFedZero under more aggressive data heterogeneity by setting $\alpha_d$ to 0.1. The results for gACC, pACC, and zACC are presented in Tabs. 7, 8, and 9, respectively. As shown, HyperFedZero continues to demonstrate strong performance in zACC, significantly outperforming all other baselines, while achieving comparable performance in gACC. Notably, HyperFedZero's personalization capability declines considerably at $\alpha_d = 0.1$, suggesting a potential trade-off between pACC and zACC, which warrants further investigation in future research.

# F    LIMITATIONS

In this work, HyperFedZero leverages a chunked-hypernetwork as its parameter generator. However, it is well-known that chunked-hypernetworks face scalability challenges, particularly when tasked with generating billions of parameters. To address this limitation, we plan to explore diffusion-based parameter generation techniques in future work. Additionally, in our supplementary experiments, we observe a trade-off between pACC and zACC performance. Specifically, as data heterogeneity increases, HyperFedZero's personalization ability (pACC) decreases significantly, while its zero-shot personalization accuracy (zACC) remains robust. This suggests a potential trade-off between optimizing zero-shot personalization accuracy and preserving personalized accuracy, which warrants further investigation in subsequent research.

Table 7: The gACC comparisons (the higher the better) between settings ($\alpha_d = 0.1$). **Bold** marks the best-performing method in each comparison.

| | MNIST | | | FMNIST | | | EMNIST | | | SVHN | | C-10 | C-100 | T-ImageNet |
|---|---|---|---|---|---|---|---|---|---|---|---|---|---|---|
| | MLP | LeNet-S | LeNet | MLP | LeNet-S | LeNet | MLP | LeNet-S | LeNet | ZekenNet | ResNet | | ResNet | |
| *N = 10* | | | | | | | | | | | | | | |
| Local | - | - | - | - | - | - | - | - | - | - | - | - | - | - |
| FedAvg | 89.79 | 94.93 | 96.35 | 82.06 | 80.86 | 83.86 | 60.53 | 74.77 | 78.01 | 78.94 | 69.59 | 28.90 | 12.55 | 6.79 |
| FedAvg-FT | 56.52 | 40.05 | 69.51 | 47.08 | 33.86 | 47.25 | 15.07 | 10.77 | 26.40 | 24.29 | 26.42 | 21.68 | 2.23 | 0.85 |
| FedProx | 89.42 | 94.65 | 96.04 | 81.93 | 79.73 | 82.87 | 59.53 | 74.57 | 77.34 | 77.23 | 71.96 | 29.01 | 12.82 | 6.90 |
| Ditto | 88.89 | 93.92 | 95.19 | 81.60 | 78.42 | 81.20 | 58.22 | 73.42 | 75.58 | 72.53 | 57.51 | 24.50 | 5.85 | 3.64 |
| Scaffold | **95.09** | 95.15 | 95.88 | 85.36 | 79.49 | 81.31 | 71.09 | 75.16 | 78.24 | 79.89 | 74.71 | 29.95 | 13.09 | 6.60 |
| pFedMe | 89.44 | 94.04 | 95.38 | 81.81 | 79.88 | 83.41 | 57.89 | 73.82 | 76.22 | 75.56 | 64.51 | 27.21 | 9.68 | 4.05 |
| pFedHN | - | - | - | - | - | - | - | - | - | - | - | - | - | - |
| PerFedAvg | 88.45 | 68.53 | 67.03 | 74.20 | 72.40 | 73.40 | 57.67 | 33.41 | 43.32 | 24.07 | 61.34 | 27.72 | 11.09 | 6.64 |
| FedAMP | 55.01 | 40.10 | 70.08 | 45.24 | 33.27 | 44.88 | 14.85 | 9.72 | 26.40 | 26.14 | 26.10 | 21.70 | 2.35 | 0.80 |
| GA | 89.69 | 95.09 | 96.14 | 81.64 | 79.96 | 82.50 | 60.16 | 75.70 | 77.72 | 78.19 | **73.24** | 27.81 | 12.96 | 6.73 |
| FedSR | 92.06 | **96.39** | 96.96 | 83.44 | 83.90 | 85.67 | 65.01 | **78.06** | **80.18** | 80.80 | 69.26 | 26.98 | 9.91 | 5.06 |
| Ensemble | 80.86 | 84.59 | 85.84 | 73.15 | 65.77 | 68.20 | 18.75 | 56.76 | 62.36 | 56.90 | 68.66 | 35.06 | 12.92 | 6.09 |
| FedJETs | 89.63 | 91.36 | 96.01 | 81.01 | 79.92 | 83.44 | 60.33 | 75.16 | 78.33 | 80.26 | 69.79 | 34.57 | 10.56 | 3.93 |
| HyperFedZero | 94.06 | 96.31 | **97.75** | **85.52** | **83.97** | **86.36** | **72.58** | 75.23 | 78.94 | **81.01** | 71.27 | **38.76** | **13.28** | **6.97** |
| *N = 50* | | | | | | | | | | | | | | |
| Local | - | - | - | - | - | - | - | - | - | - | - | - | - | - |
| FedAvg | 91.17 | 94.24 | 97.32 | 82.34 | 81.53 | 83.79 | 64.32 | 78.56 | 80.49 | 82.28 | 75.37 | 35.74 | 15.80 | 6.95 |
| FedAvg-FT | 61.84 | 36.91 | 63.24 | 34.26 | 32.80 | 46.83 | 18.91 | 9.13 | 32.29 | 25.35 | 24.26 | 21.44 | 2.15 | 0.76 |
| FedProx | 90.69 | 5.29 | 97.13 | 81.72 | 79.99 | 82.80 | 62.95 | 78.17 | 80.08 | 81.87 | 76.85 | 36.03 | 16.25 | 7.28 |
| Ditto | 89.66 | 93.44 | 96.17 | 80.75 | 77.49 | 79.67 | 61.85 | 74.22 | 77.79 | 77.18 | 63.60 | 27.16 | 4.46 | 1.62 |
| Scaffold | 92.74 | 93.75 | **98.23** | 83.02 | 80.49 | 81.56 | 68.50 | **80.53** | 81.96 | 74.26 | 71.76 | 25.04 | **21.65** | **11.43** |
| pFedMe | 90.56 | 96.19 | 96.56 | 81.62 | 80.06 | 81.97 | 61.16 | 75.44 | 77.73 | 80.57 | 71.67 | 31.41 | 11.34 | 3.22 |
| pFedHN | - | - | - | - | - | - | - | - | - | - | - | - | - | - |
| PerFedAvg | 91.06 | 79.44 | 92.04 | 77.60 | 48.90 | 67.40 | 63.37 | 78.11 | 80.27 | 79.76 | 70.73 | 32.56 | 16.15 | 6.94 |
| FedAMP | 60.34 | 36.53 | 61.16 | 34.81 | 32.91 | 45.76 | 19.34 | 9.59 | 32.63 | 24.45 | 23.70 | 21.69 | 2.20 | 0.75 |
| GA | 90.49 | 96.32 | 96.92 | 81.04 | 78.29 | 80.95 | 63.06 | 78.50 | 80.69 | 80.53 | **78.46** | 36.45 | 16.49 | 7.15 |
| FedSR | 91.92 | 96.06 | 98.12 | 83.87 | 84.35 | 85.13 | 66.37 | 80.22 | **82.16** | **83.29** | 76.20 | 33.86 | 12.70 | 4.77 |
| Ensemble | 86.33 | 90.54 | 91.38 | 74.82 | 66.75 | 68.38 | 13.64 | 61.25 | 66.25 | 59.02 | 76.30 | **42.73** | 15.92 | 4.93 |
| FedJETs | 91.86 | 95.22 | 97.34 | 83.61 | 84.33 | 81.97 | 64.13 | 75.71 | 80.13 | 39.01 | 74.52 | 38.94 | 15.33 | 5.51 |
| HyperFedZero | **94.22** | **96.79** | 97.97 | **84.62** | **84.63** | **86.77** | **70.98** | 76.49 | 80.34 | 82.49 | 74.56 | 40.84 | 12.71 | 5.66 |

# G  DISCLOSURE OF LLM USAGE

LLMs were used to aid in writing and polishing the text of this paper. All content has been reviewed by the authors, who take full responsibility for the work.

Table 8: The pACC comparisons (the higher the better) between settings ($\alpha_d = 0.1$). **Bold** marks the best-performing method in each comparison.

| | MNIST MLP | MNIST LeNet-S | MNIST LeNet | FMNIST MLP | FMNIST LeNet-S | FMNIST LeNet | EMNIST MLP | EMNIST LeNet-S | EMNIST LeNet | SVHN ZekenNet | SVHN ResNet | C-10 | C-100 ResNet | T-ImageNet ResNet |
|---|---|---|---|---|---|---|---|---|---|---|---|---|---|---|
| | | | | | | | | $N = 10$ | | | | | | |
| Local | 97.15 | 98.41 | 98.47 | 93.81 | 94.46 | 94.57 | 85.88 | 90.37 | **91.33** | 85.49 | 73.53 | **84.92** | 25.48 | 10.49 |
| FedAvg | 88.36 | 94.05 | 95.21 | 83.22 | 82.08 | 83.99 | 63.10 | 78.25 | 81.31 | 81.72 | 59.04 | 30.94 | 12.53 | 6.66 |
| FedAvg-FT | **97.15** | 98.41 | 98.47 | 93.81 | 94.48 | **94.62** | **85.88** | 90.34 | 91.22 | **85.73** | 74.02 | 84.81 | 24.49 | 10.20 |
| FedProx | 87.92 | 93.62 | 94.92 | 82.93 | 80.91 | 83.23 | 62.34 | 78.24 | 81.03 | 80.27 | 62.67 | 31.76 | 13.21 | 7.14 |
| Ditto | 87.58 | 92.51 | 93.99 | 82.65 | 79.88 | 82.12 | 61.76 | 76.74 | 79.04 | 76.26 | 49.22 | 19.21 | 5.47 | 3.31 |
| Scaffold | 94.91 | 95.60 | 94.42 | 85.82 | 80.57 | 82.38 | 75.11 | 78.90 | 81.64 | 80.66 | **77.51** | 31.11 | 13.06 | 7.48 |
| pFedMe | 87.84 | 92.99 | 94.11 | 82.56 | 81.01 | 83.84 | 61.27 | 77.73 | 80.40 | 78.75 | 56.16 | 27.67 | 9.99 | 4.19 |
| pFedHN | 96.45 | 95.97 | 97.98 | 92.66 | 91.06 | 93.01 | 81.94 | 78.56 | 86.67 | 80.75 | 71.65 | 82.62 | **28.92** | **16.62** |
| PerFedAvg | 86.66 | 64.24 | 62.64 | 73.96 | 74.26 | 73.43 | 60.67 | 34.82 | 44.85 | 23.75 | 48.94 | 21.19 | 10.27 | 6.86 |
| FedAMP | 97.03 | **98.46** | **98.47** | **93.84** | **94.55** | 94.52 | 85.81 | **90.57** | 91.16 | 85.58 | 73.79 | 85.11 | 24.67 | 10.07 |
| GA | 89.68 | 94.78 | 95.47 | 83.02 | 81.30 | 83.51 | 63.90 | 79.65 | 82.00 | 82.07 | 66.56 | 32.47 | 13.01 | 6.83 |
| FedSR | 90.92 | 96.11 | 96.31 | 84.07 | 84.15 | 86.60 | 68.15 | 80.77 | 82.81 | 84.53 | 59.69 | 34.31 | 10.42 | 4.95 |
| Ensemble | 76.77 | 81.15 | 83.64 | 73.62 | 66.55 | 69.72 | 16.40 | 59.79 | 66.35 | 58.49 | 61.26 | 33.43 | 12.87 | 6.82 |
| FedJETs | 89.01 | 90.11 | 94.90 | 82.30 | 80.55 | 84.31 | 63.61 | 76.90 | 79.87 | 83.88 | 66.60 | 39.73 | 9.95 | 3.87 |
| HyperFedZero | 93.46 | 95.80 | 97.13 | 85.77 | 84.51 | 86.50 | 75.13 | 77.88 | 81.85 | 83.57 | 74.76 | 46.80 | 13.66 | 7.24 |
| | | | | | | | | $N = 50$ | | | | | | |
| Local | 89.95 | 96.96 | 97.06 | 92.75 | 93.52 | 93.74 | 82.02 | **86.21** | 86.34 | 80.36 | 73.50 | 72.65 | 27.08 | 12.53 |
| FedAvg | 91.95 | 95.25 | 97.49 | 80.24 | 82.03 | 83.05 | 64.21 | 79.00 | 81.50 | 81.65 | 58.50 | 26.47 | 13.77 | 6.40 |
| FedAvg-FT | 95.78 | **96.96** | 97.06 | **92.75** | 93.66 | 93.75 | 82.02 | 86.18 | 86.38 | 80.54 | 73.27 | 72.48 | **27.89** | **12.62** |
| FedProx | 91.20 | 95.98 | 97.35 | 80.00 | 79.70 | 82.11 | 62.96 | 78.86 | 81.54 | 81.19 | 57.24 | 25.26 | 13.51 | 6.11 |
| Ditto | 90.54 | 94.72 | 96.23 | 79.45 | 76.83 | 78.93 | 62.04 | 75.28 | 78.92 | 76.44 | 42.02 | 18.84 | 3.55 | 1.73 |
| Scaffold | 93.42 | 94.74 | 98.46 | 81.37 | 80.85 | 79.85 | 68.65 | 81.95 | 83.44 | 75.20 | 73.43 | 20.72 | 18.68 | 10.19 |
| pFedMe | 91.19 | 94.75 | 96.83 | 80.02 | 80.05 | 80.57 | 61.02 | 76.59 | 78.90 | 78.95 | 51.19 | 23.29 | 9.93 | 2.69 |
| pFedHN | 93.03 | 80.29 | 93.21 | 87.40 | 72.12 | 84.67 | 71.25 | 39.41 | 79.17 | 81.38 | 61.12 | 54.83 | 19.50 | 11.97 |
| PerFedAvg | 91.75 | 78.84 | 92.97 | 77.04 | 52.16 | 66.40 | 63.14 | 78.87 | 81.00 | 76.58 | 48.04 | 22.50 | 13.76 | 5.73 |
| FedAMP | **95.79** | 96.76 | 97.07 | 75.55 | **93.80** | **93.83** | **82.05** | 86.15 | **86.43** | 80.23 | **73.44** | **72.71** | 27.22 | 12.14 |
| GA | 91.37 | 94.75 | 97.18 | 79.84 | 78.33 | 80.03 | 63.18 | 79.43 | 81.96 | 80.48 | 58.53 | 26.21 | 13.57 | 6.30 |
| FedSR | 91.80 | 96.41 | **98.63** | 82.29 | 83.74 | 83.37 | 66.16 | 81.30 | 83.14 | **82.18** | 60.39 | 26.70 | 10.85 | 4.27 |
| Ensemble | 86.26 | 90.56 | 92.24 | 74.24 | 66.89 | 69.81 | 13.06 | 62.04 | 67.01 | 55.76 | 50.30 | 30.85 | 14.00 | 4.83 |
| FedJETs | 92.35 | 95.29 | 97.40 | 82.15 | 83.24 | 80.26 | 65.40 | 75.48 | 81.19 | 36.84 | 68.05 | 32.22 | 13.45 | 5.04 |
| HyperFedZero | 94.23 | 96.59 | 98.33 | 83.09 | 84.65 | 84.67 | 71.70 | 77.35 | 81.87 | 81.38 | 73.21 | 38.19 | 12.03 | 6.51 |

Table 9: The zACC comparisons (the higher the better) between settings ($\alpha_d = 0.1$). **Bold** marks the best-performing method in each comparison.

| | MNIST MLP | LeNet-S | LeNet | FMNIST MLP | LeNet-S | LeNet | EMNIST MLP | LeNet-S | LeNet | SVHN ZekenNet | ResNet | C-10 ResNet | C-100 ResNet | T-ImageNet ResNet |
|---|---|---|---|---|---|---|---|---|---|---|---|---|---|---|
| | | | | | | | $N=10$ | | | | | | | |
| Local | 2.40 | 1.56 | 0.96 | 4.43 | 1.30 | 0.39 | 0.46 | 0.09 | 3.95 | 51.37 | 7.75 | 0.00 | 0.00 | 0.08 |
| FedAvg | 94.47 | 98.08 | 97.84 | 94.79 | 94.40 | 95.70 | 31.71 | 49.91 | 54.41 | 57.10 | 41.60 | 7.68 | 5.52 | 3.13 |
| FedAvg-FT | 87.02 | 66.11 | 89.06 | 89.58 | 73.31 | 73.31 | 5.42 | 0.37 | 9.10 | 7.49 | 13.74 | 7.95 | 0.42 | 0.63 |
| FedProx | 94.35 | 97.36 | 97.60 | 94.66 | 94.14 | 95.83 | 30.15 | 49.36 | 54.96 | 53.39 | 45.05 | 7.63 | 6.04 | 3.05 |
| Ditto | 94.11 | 97.36 | 97.36 | 94.66 | 94.79 | 95.44 | 30.79 | 46.97 | 51.38 | 45.83 | 32.94 | 6.48 | 3.33 | 1.56 |
| Scaffold | 95.55 | 96.39 | 96.51 | 94.92 | 93.75 | 95.31 | 36.40 | 47.15 | 52.85 | 60.03 | 44.47 | 10.75 | 6.46 | 1.89 |
| pFedMe | 94.35 | 97.24 | 97.48 | 94.79 | 94.14 | 95.83 | 29.96 | 47.89 | 53.31 | 53.26 | 39.00 | 7.08 | 4.58 | 1.80 |
| pFedHN | 26.08 | 48.20 | 10.70 | 8.07 | 0.52 | 2.47 | 5.33 | 1.84 | 0.64 | 6.19 | 0.20 | 0.05 | 0.10 | 0.00 |
| PerFedAvg | 94.23 | 89.66 | 91.11 | 93.36 | 91.41 | 91.93 | 33.00 | 13.51 | 26.75 | 9.83 | 31.25 | 11.76 | 4.58 | 3.20 |
| FedAMP | 86.78 | 69.47 | 86.66 | 89.32 | 68.75 | 71.48 | 5.61 | 0.28 | 8.82 | 9.25 | 13.09 | 7.95 | 0.42 | 0.55 |
| GA | 94.47 | 97.72 | 97.60 | 95.18 | 94.92 | 96.35 | 36.31 | 51.65 | 55.53 | 55.40 | 44.34 | 9.74 | 7.50 | 3.20 |
| FedSR | 95.91 | 98.56 | 97.96 | 93.75 | 94.53 | 95.83 | 33.64 | 51.56 | 53.22 | 57.16 | 40.04 | 6.80 | 5.42 | 2.11 |
| Ensemble | 82.69 | 96.03 | 95.19 | 87.11 | 88.54 | 89.19 | 0.46 | 34.56 | 37.22 | 25.39 | 42.45 | 6.89 | 5.10 | 1.95 |
| FedJETs | 93.03 | 94.47 | 98.08 | 92.58 | 89.58 | 93.88 | 32.90 | 51.38 | 55.70 | 60.61 | 45.51 | 8.36 | 5.72 | 1.56 |
| HyperFedZero | **96.39** | **98.72** | **98.68** | **95.23** | **95.57** | **96.48** | **50.49** | **52.02** | **55.97** | **60.81** | **48.24** | **16.59** | **9.90** | **4.84** |
| | | | | | | | $N=50$ | | | | | | | |
| Local | 4.68 | 11.11 | 3.47 | 0.00 | 2.77 | 33.33 | 0.00 | 0.69 | 4.86 | 1.50 | 8.27 | 0.00 | 0.78 | 0.34 |
| FedAvg | 89.58 | 92.36 | 96.52 | 82.63 | 65.27 | 77.08 | 62.50 | 70.13 | 74.30 | 75.93 | 54.88 | 11.45 | 7.03 | 3.12 |
| FedAvg-FT | 60.41 | 6.25 | 63.88 | 24.30 | 2.77 | 2.08 | 4.16 | 7.63 | 28.47 | 44.36 | 43.60 | 1.56 | 3.90 | 0.34 |
| FedProx | 88.19 | 93.75 | 96.52 | 78.47 | 63.88 | 74.30 | 60.41 | 70.83 | 74.30 | 72.93 | 58.64 | 12.50 | 7.81 | 3.47 |
| Ditto | 87.50 | 92.36 | 97.91 | 79.86 | 63.88 | 65.27 | 56.94 | 71.52 | 73.61 | 69.92 | 54.13 | 4.68 | 2.34 | 1.38 |
| Scaffold | 90.97 | 91.66 | 98.61 | 81.25 | 65.97 | 77.08 | 64.53 | 71.52 | 74.30 | 73.68 | 69.17 | 11.04 | 10.93 | 3.12 |
| pFedMe | 89.58 | 93.05 | 96.52 | 79.86 | 67.36 | 70.13 | 56.94 | 69.44 | 72.91 | 72.18 | 66.91 | 9.89 | 5.46 | 2.43 |
| pFedHN | 42.36 | 2.08 | 4.16 | 22.22 | 41.66 | 85.41 | 26.38 | 1.38 | 1.38 | 71.42 | 66.91 | 0.50 | 1.56 | 1.38 |
| PerFedAvg | 88.88 | 70.83 | 81.25 | 75.69 | 27.77 | 59.02 | 65.27 | 70.83 | 74.30 | 76.69 | 71.42 | 13.02 | 7.81 | 4.16 |
| FedAMP | 53.47 | 5.55 | 61.80 | 21.52 | 2.77 | 4.16 | 4.16 | 9.02 | 27.77 | 53.38 | 63.90 | 1.56 | 3.90 | 0.34 |
| GA | 88.88 | 93.05 | 98.61 | 79.86 | 68.75 | 72.22 | 56.94 | 70.13 | 70.13 | 76.69 | 64.66 | 10.93 | 8.59 | 3.81 |
| FedSR | 90.97 | 94.44 | 95.13 | 83.33 | 77.08 | 80.55 | 64.53 | 69.44 | 74.30 | 75.93 | 71.42 | 16.14 | 9.37 | 3.47 |
| Ensemble | 86.11 | 86.80 | 84.02 | 70.83 | 44.44 | 45.13 | 4.16 | 61.80 | 63.88 | 51.12 | 68.42 | 13.02 | 10.15 | 2.43 |
| FedJETs | 90.27 | 93.75 | 81.94 | 74.30 | 80.55 | 81.25 | 63.88 | 70.13 | 76.38 | 66.16 | 75.93 | 23.54 | 12.50 | 4.16 |
| HyperFedZero | **92.36** | **95.13** | **99.30** | **85.41** | **85.41** | **88.89** | **68.05** | **72.22** | **77.78** | **78.94** | **77.44** | **42.18** | **14.84** | **6.86** |

