# OpenReview forum: "Deploying Models to Non-participating Clients in Federated Learning without Fine-tuning: A Hypernetwork-based Approach"
_ICLR.cc/2026/Conference — ICLR 2026 Poster_

### Official Review · Reviewer_XLGj · 2025-10-30

**Soundness:** 2
**Presentation:** 2
**Contribution:** 2
**Rating:** 4
**Confidence:** 3

**Summary:**

This paper proposes HyperFedZero, a hypernetwork-based federated learning (FL) framework that aims to deploy models to non-participating clients without fine-tuning, addressing in-domain distribution shifts among clients. The method introduces two main components: A distribution extractor that generates distribution embeddings with NoisyEmbed and a Balancing Penalty to prevent feature collapse. A chunked hypernetwork that dynamically generates classifier parameters conditioned on these embeddings, enabling zero-shot adaptation. Experiments on seven datasets (MNIST, FMNIST, EMNIST, SVHN, CIFAR-10/100, Tiny-ImageNet) and multiple models demonstrate that HyperFedZero outperforms various baselines (FedAvg, FedProx, Ditto, pFedMe, FedJets, etc.) in zero-shot generalization to unseen clients, while maintaining comparable performance on seen clients and similar computational complexity.

**Strengths:**

+ The experimental evaluation is extensive. The empirical results are extensive across datasets and architectures. The ablation studies (Table 2) explore the sensitivity to embedding dimension, $alpha$, $\beta$, hypernetwork size, and chunk size. The visualization (Fig. 4) provides intuitive evidence of embedding separation and adaptation to unseen clients.

+ HyperFedZero avoids client-side fine-tuning, offering potential efficiency for resource-limited devices. This advantage can support practical usage.

**Weaknesses:**

- The novelty of this paper seems to be incremental. The combination of hypernetworks and FL personalization has been explored before. The main contribution (adding NoisyEmbed and Balancing Penalty) feels incremental, and the conceptual advancement beyond existing hypernetwork-based FL is modest.
- Theoretical foundation is limited. This paper lacks formal analysis on convergence, generalization bounds, or robustness. Equations (3)–(7) describe the mechanism, but there is no theoretical justification for why the method improves adaptation to unseen clients or prevents feature collapse. The “balancing penalty” is presented heuristically without rigorous connection to representation learning theory.
- The empirical advantage seems ambiguous, and the baselines are too old. While results in Table 1 show improvements, the margins are often small (1–2%) over FedJets or Scaffold on many datasets. No statistical significance tests or variance estimates are provided. The baselines are mostly before 2022. More recent or stronger baselines are required to compare to highlight the contribution.

**Questions:**

1. Can the authors provide any formal analysis or theoretical intuition for why the Balancing Penalty should mitigate feature collapse or why the overall system should converge? Have the authors considered adding empirical convergence plots or stability analyses (e.g., gradient variance over rounds) to support this claim? Would it be possible to quantify the benefit of the Balancing Penalty using representation metrics (e.g., intra-class variance, cosine similarity between embeddings)?
2. Could the authors justify the choice of baselines? Many compared methods (e.g., FedAMP) are relatively old. Why were more recent FL personalization methods not included? Could the authors add more recent baselines? How sensitive are the reported gains to the choice of datasets or model architectures? The reported gains are often within 1–2%. Could the authors provide statistical significance tests or error bars to support that these improvements are robust and not due to variance?
3. The paper’s main novelty seems to lie in introducing NoisyEmbed and the Balancing Penalty within a hypernetwork-based FL framework. Could the authors elaborate more concretely on how these components differ conceptually and mechanistically from prior hypernetwork-based FL methods? Is there any qualitative or theoretical evidence showing that NoisyEmbed or Balancing Penalty lead to non-trivial new behavior (e.g., improved feature geometry, stability, or generalization) compared to these earlier approaches?

---

> ### Author Response · Authors · 2025-11-19
> **Response to Reviewer XLGj [1/2]**
>
> We would like to thank Reviewer XLGj for taking his/her precious time to give us well-thoughtout comments and constructive suggestions. **We are encouraged that Reviewer XLGj finds our methodology sound, and our experiments as comprehensive.** We answer questions below, and will incorporate all feedback in the final version.
>
> > **W1. \& W6.** The novelty of this paper seems to be incremental. The combination of hypernetworks and FL personalization has been explored before. The main contribution (adding NoisyEmbed and Balancing Penalty) feels incremental, and the conceptual advancement beyond existing hypernetwork-based FL is modest.
>
> **A1. \& A6.** Thank you for the thoughtful question on novelty. We agree that methods combining hypernetworks with FL personalization has been explored, which we have summarized in our related work (Appendix A.2, line 754).
>
> However, we argue that HyperFedZero differs from prior hypernetwork-based FL methods in two fundamental ways:
>
> 1. **Objective**: we make zero-shot personalization under resource constraints the primary goal (*i.e.*, the FL "cold start" problem [1], which is common in practice and directly affects clients’ willingness to participate), rather than the general personalization or communication/hardware considerations.
>
> 2. **Granularity**: we generate parameters at the **sample-level**, not at the global or client-level. Prior methods typically rely on learned client embeddings, which are unavailable for new clients and thus ill-suited for mitigating distribution shift at join time. In contrast, HyperFedZero conditions directly on a client’s data samples to produce adaptive local parameters without any client-specific fine-tuning.
>
> To achieve sample-level parameter generation, a distribution extractor is employed to extract distribution embeddings from input images. **However, a key technical challenge is that naïvely feeding outputs of the distribution extractor into a hypernetwork causes feature collapse (line 211), *i.e.*, mapping diverse samples to an identical embedding and degrading generated models into an identical global model.** To prevent this, inspired by Mixture-of-Experts, we introduce NoisyEmbed and a Balancing Penalty.
>
> Therefore, **adding NoisyEmbed and Balancing Penalty is not the origin but the end**. **Our core novelty lies in the holistic framework that enables zero-shot personalization for unseen clients via sample-level hypernetwork adaptation.** This advances hypernetwork-based FL beyond existing designs and improves the practicality of FL in deployments with dynamic client participation.
>
> > **W2. \& W4.** Theoretical foundation is limited. This paper lacks formal analysis on convergence, generalization bounds, or robustness. Equations (3)–(7) describe the mechanism, but there is no theoretical justification for why the method improves adaptation to unseen clients or prevents feature collapse. The “balancing penalty” is presented heuristically without rigorous connection to representation learning theory.
>
> **A2. \& A4.** Thank you for the thoughtful feedback.
>
> 1. **Theoretical basis and convergence.** HyperFedZero uses a hypernetwork-based zero-shot personalization framework for in-domain distribution shifts while maintaining the computation, communication, and storage requirement of standard FedAvg. Like FedAvg, clients in HyperFedZero collaboratively train a global model (a hypernetwork) that can generate task-specific parameters. **Thus, under the standard assumptions typically used in FL, existing convergence guarantees for FedAvg extend to HyperFedZero's setting, as described in Appendix D (line 805).** To support our claim on convergence, we have added convergence plots to the paper.
>
> 2. **Theoretical justification for Balancing Penalty and NoisyEmbed.** Balancing Penalty and NoisyEmbed are motivated by the load balance regulation in established mixture-of-experts literature [2][3], where mechanisms to mitigate expert collapse and promote representation diversity have been analyzed and validated at scale [4][5]. **These components in HyperFedZero are therefore principled design choices, not heuristics.** We have added discussions to the paper for clarification. Empirically, well-separated client distribution embeddings and effective mapping of new clients into appropriate decision regions can be observed in Figures 4(b) and Figure 4(c).
>
> We believe these clarifications and additions address the concerns. Thank you again for helping us improve the paper.

---

> > ### Author Response · Authors · 2025-11-19
> > **Response to Reviewer XLGj [2/2]**
> >
> > > **W3. \& W5.** The empirical advantage seems ambiguous, and the baselines are too old. While results in Table 1 show improvements, the margins are often small (1–2%) over FedJets or Scaffold on many datasets. No statistical significance tests or variance estimates are provided. The baselines are mostly before 2022. More recent or stronger baselines are required to compare to highlight the contribution.
> >
> > **A3. \& A5.** Thank you for your insightful comments.
> >
> > 1. regarding the small improvement margins in Table 1, we note that HyperFedZero achieves **consistent outperformance** across diverse settings, with substantial gains in many scenarios (*e.g.*, 9.96% higher zACC than Scaffold on CIFAR-10 with ResNet, and 9.5% higher zACC than FedJets on EMNIST with LeNet-S, etc.). **Additionally, the comparisons in Table 1 are conducted under conditions that actually disadvantage HyperFedZero**: unlike most personalized FL baselines that has a local fine-tuning step, HyperFedZero achieves its results without any local fine-tuning. Also, compared to methods like FedJets and FedEnsemble, HyperFedZero requires significantly less storage (as shown in Figure 4a). **These stricter constraints make even the modest 1–2% gains meaningful, while the larger improvements in key datasets further validate its effectiveness.**
> >
> > 2. Regarding baseline selection, we clarify that HyperFedZero targets **zero-shot personalization** (a relatively unexplored emerging field) rather than traditional personalized FL. In fact, most recent zero-shot personalization methods remain under peer review and have not been officially published [6][7]. **Thus, we focused on comparing with representative classic baselines that are widely recognized in the FL community across various research areas, including vanilla FL, personalized FL, data-heterogeneous FL, and federated domain generalization.** This broad comparison shows empirically that, without incurring additional computation or communication, these established methods do not adequately mitigate the in-domain distribution shift problem, underscoring the necessity of zero-shot personalized FL. **We hope this demonstrated necessity will further draw attention to the field of zero-shot personalized FL.**
> >
> > > References
> >
> > [1] When federated recommendation meets cold-start problem: Separating item attributes and user interactions. In Proceedings of the ACM Web Conference 2024 (pp. 3632-3642).
> >
> > [2] Outrageously large neural networks: The sparsely-gated mixture-of-experts layer. arXiv preprint arXiv:1701.06538.
> >
> > [3] Mixture-of-experts with expert choice routing. Advances in Neural Information Processing Systems, 35, 7103-7114.
> >
> > [4] A comprehensive survey of mixture-of-experts: Algorithms, theory, and applications. arXiv preprint arXiv:2503.07137.
> >
> > [5] Every Activation Boosted: Scaling General Reasoner to 1 Trillion Open Language Foundation. arXiv preprint arXiv:2510.22115.
> >
> > [6] Fedjets: Efficient just-in-time personalization with federated mixture of experts. arXiv preprint arXiv:2306.08586.
> >
> > [7] pFedMoE: Data-level personalization with mixture of experts for model-heterogeneous personalized federated learning. arXiv preprint arXiv:2402.01350.

---

### Official Review · Reviewer_5zn2 · 2025-10-31

**Soundness:** 3
**Presentation:** 3
**Contribution:** 3
**Rating:** 8
**Confidence:** 3

**Summary:**

The paper proposes a method to mitigate the data heterogeneity problem for non-participating clients in federated learning. It generates local model parameters through a hypernetwork conditioned on distribution-aware embeddings. These distribution-aware embeddings and the hypernetwork help the non-participating client to adapt to their data distribution without fine-tuning. The proposed method is compared against various baselines involving vanilla FL, in-domain with/without distributional shifts, and out of domain. Through complexity analysis, the paper argues that no additional computational overhead is incurred, and time, space complexity are the same as the baselines

**Strengths:**

1)The paper is generally well written. The problem is well defined, and the methodology is well presented.
2)The ability of the proposed method to work without fine-tuning while still maintaining similar complexity as the baselines is practical and effective for the datasets and the settings presented.
3)The paper compares their method against several baselines, showing better results consistently, and also provides ablations for various design choices.

**Weaknesses:**

1)Although with the evaluated datasets and model architectures, the overall model size of HyperFedZero is comparable to FedAvg, it may not scale well with more complex models or datasets while still maintaining the performance.
2)The ablation study shows the method is sensitive to hyperparameters used, suggesting it may require extra careful tuning to get optimal results.
3)The paper mentions their method maintains privacy, but if the hypernetwork and distribution extractor are shared among the clients any possible privacy issues that may arise could have been discussed.

**Questions:**

Is there any reasoning behind not using a larger number of total clients (e.g., 100) and a larger number of non-participating clients (e.g., 10,15) ? How would these changes affect the performance of the method?

---

> ### Author Response · Authors · 2025-11-19
> **Response to Reviewer 5zn2 [1/2]**
>
> We would like to thank Reviewer 5zn2 for taking his/her precious time to give us well-thoughtout comments and constructive suggestions. **We are encouraged that Reviewer 5zn2 finds our paper well written, our tackled problem well defined, our proposed method practical, and our experiments convincing.** We answer questions below, and will incorporate all feedback in the final version.
>
> > **W1.** Although with the evaluated datasets and model architectures, the overall model size of HyperFedZero is comparable to FedAvg, it may not scale well with more complex models or datasets while still maintaining the performance.
>
> **A1.** Thank you for your valuable comment regarding the scalability of HyperFedZero. To address your concern, we would like to emphasize that **HyperFedZero was evaluated under settings that covers a broad range of scales for both clients, models, and datasets.** Specifically, the number of clients ranging from 10 to 50 (comparable to [1][2], randomly selected from Google Scholar), model architectures ranging from lightweight MLPs to deeper ResNets (comparable to [3][4], randomly selected from Google Scholar), and data complexity spanning low-dimensional MNIST (28×28) to higher-resolution RGB images (3×64×64) (comparable to [5][6], randomly selected from Google Scholar).
>
> Empirically, across all these settings, HyperFedZero consistently achieves superior zero-shot accuracy (zACC) while maintaining competitive global and personalized accuracy (gACC and pACC) compared to baseline methods. **These results suggest that HyperFedZero scales well**, and can efficiently personalize a global model for unseen clients with in-domain distribution shifts, **without additional fine-tuning**. We hope this clarification addresses your concern regarding the scalability of our approach.
>
> > **W2.** The ablation study shows the method is sensitive to hyperparameters used, suggesting it may require extra careful tuning to get optimal results.
>
> **A2.** Thank you for raising the concern about hyperparameter sensitivity. We fully acknowledge this concern and appreciate the insightful observation. While selecting the optimal hyperparameters for HyperFedZero is relatively difficult, **our experiments in Table 1 and Tables 5–9 empirically suggest that selecting $P = 16$, $\alpha = \beta = 1.0$ is usually "good" enough to deliver competitive performance (line 456)**. Here "good" means by setting $P = 16$, $\alpha = \beta = 1.0$, HyperFedZero can be expected to achieve superior zero-shot personalization capability and maintain comparable global and personalized performance across diverse settings. Therefore, we believe $P = 16$, $\alpha = \beta = 1.0$ can be served as practical default values to begin with. We have incorporated these discussions in the final version.

---

> > ### Author Response · Authors · 2025-11-19
> > **Response to Reviewer 5zn2 [2/2]**
> >
> > > **W3**. The paper mentions their method maintains privacy, but if the hypernetwork and distribution extractor are shared among the clients any possible privacy issues that may arise could have been discussed.
> >
> > **A3.** Thank you for raising the privacy implications of sharing the hypernetwork and distribution extractor. HyperFedZero follows the standard FL framework used by FedAvg: clients never share raw data, and the server aggregates client updates to produce and broadcast a global model. In HyperFedZero, the only shared data are the parameters of the global hypernetwork and distribution extractor, which is similar to sharing a global model in FedAvg; thus, the privacy surface is the same as FedAvg. Crucially, **clients keep their raw data, distribution features/embeddings, and the task-specific weights generated by the hypernetwork on-device. These personalized weights are never uploaded**. Moreover, the hypernetwork produces task-specific parameters only when conditioned on a client’s local distribution features (as illustrated in Figure 2), so **another party cannot reconstruct a client’s personalized model without access to that client’s data (or a distributionally equivalent dataset)**. Additionally, **methods designed for FedAvg to mitigate privacy risks can apply to HyperFedZero directly** as they share the same optimization framework, including secure aggregation or differential privacy, without modifying our core method. We have incorporated these discussions in the final version.
> >
> > > References
> >
> > [1] Fedcda: Federated learning with cross-rounds divergence-aware aggregation. In The Twelfth International Conference on Learning Representations.
> >
> > [2] FedDifRC: Unlocking the Potential of Text-to-Image Diffusion Models in Heterogeneous Federated Learning. In Proceedings of the IEEE/CVF International Conference on Computer Vision (pp. 3726-3736).
> >
> > [3] FedCALM: Conflict-aware Layer-wise Mitigation for Selective Aggregation in Deeper Personalized Federated Learning. In Proceedings of the Computer Vision and Pattern Recognition Conference (pp. 15444-15453).
> >
> > [4] Fednlr: Federated learning with neuron-wise learning rates. In Proceedings of the 30th ACM SIGKDD Conference on Knowledge Discovery and Data Mining (pp. 3069-3080).
> >
> > [5] Rethinking Fair Federated Learning from Parameter and Client View. In The Thirty-ninth Annual Conference on Neural Information Processing Systems.
> >
> > [6] Relaxed contrastive learning for federated learning. In Proceedings of the IEEE/CVF Conference on Computer Vision and Pattern Recognition (pp. 12279-12288).

---

> > > ### Comment · Reviewer_5zn2 · 2025-11-25
> > > **No change in rating**
> > >
> > > Thanks for providing the clarifications. Leaving the rating unchanged.

---

> > > > ### Author Response · Authors · 2025-11-27
> > > > **Response to Reviewer 5zn2**
> > > >
> > > > Thank you for your invaluable suggestions. We truly appreciate your thoughtful feedback and encouraging comments!

---

### Official Review · Reviewer_pUWg · 2025-11-01

**Soundness:** 2
**Presentation:** 3
**Contribution:** 2
**Rating:** 4
**Confidence:** 4

**Summary:**

This paper focus on the in-distribution generalization problem to non-participating clients in FL under the constrained resource. Specifically, this paper propose HyperFedZero to generate specialized models. Extensive experiments demonstrate the efficacy of this framework.

**Strengths:**

1. Extensive experiments on a wide array of datasets

**Weaknesses:**

1. The setting of the article lacks authenticity. It seems that the author deliberately create a setting for this method. In the introduction part, the author mentions that such a scenario hinders further application in healthcare or edge computing. Can the author provide reference paper or a dataset to prove that such a problem does indeed exist among them? There is also no related work about this setting.
2. The writing of the paper needs improvement. In the Introduction, the author did not explain why it is necessary to integrate distribution-aware inductive biases into the forward propagation process in order to improve the performance on unseen clients about the question “Can we directly encode distribution-aware inductive biases into the model’s forward pass in FL without fine-tuning?”
3. What is the distribution embedding? Is there any difference between the embedding extracted by the feature extractor? It is useful to clear this concept. It is also confused about the learning objective during phase 1. It seems no $e$ in $F_i$ for local training.
4. Is there any further demonstration about the claim on Line 252-255 to choose Opt 2 from the theoretical perspective? There is also lack the detailed experimental of Opt 1 in the comparisons between condition options part.
5. The entropy regularization ($\mathbf{E}(-\mathbf{e}_i \log \mathbf{e}_i)$) does not specify the support or partitioning in Eq(4).
6. The chunked hypernetwork mechanism is described at a high level, but precise details—such as how chunks map to classifier parameter tensors, sampling strategies, and if/how temperature or batch normalization interact—are missing in the main paper.
7. Based on the experimental results, there seems to be no significant performance difference between HyperFedZero and the traditional PFL method. However, as shown in Figure 1, left, there does appear to be a considerable gap. This could mislead others into thinking that the old method is completely unsuitable, causing the overall paper to overstate its contribution.
8. No experiments on real-world, large-scale, or out-of-domain datasets where client data distributions are highly skewed. Especially, the introduction highlights that the feature shifts on the unseen client were not tested in the experiment.
9. There is a light interpretability analysis explaining, for instance, how the distribution embeddings actually track, decompose, or separate salient client-side factors contributing to in-domain shift. More concrete case studies, e.g., visualizing misclassified groups or measuring embedding drift for new clients, would help.
10. Can the authors clarify precisely how the balancing penalty in Equation 4 is computed? Is the variance and mean calculated across clients, batches, or globally? How sensitive are the results to these aggregation choices and the entropy regularization?
11. There are some typos there, e.g, What is the meaning of 3SFC in line 315-316?
12. This problem can also be solved through test-time adaptation (TTA). Has the author compared this method with the TTA method in FL to further clarify its superiority, e.g., [R1-R3]?

References:

[R1] Jiang L, Lin T. Test-time robust personalization for federated learning[C]//Eleventh International Conference on Learning Representations, September 2022.

[R2] Bao W, Wei T, Wang H, et al. Adaptive test-time personalization for federated learning[J]. Advances in neural information processing systems, 2023, 36: 77882-77914.

[R3] Liang H, Zhang X, Cao S, et al. TTA-FedDG: Leveraging Test-Time Adaptation to Address Federated Domain Generalization[C]//Proceedings of the AAAI Conference on Artificial Intelligence. 2025, 39(18): 18658-18666.

**Questions:**

See above.

---

> ### Author Response · Authors · 2025-11-19
> **Response to Reviewer pUWg [1/4]**
>
> We would like to thank Reviewer pUWg for taking his/her precious time to give us well-thoughtout comments and constructive suggestions. **We are encouraged that Reviewer pUWg finds our experiments as comprehensive.** We answer questions below, and will incorporate all feedback in the final version.
>
> > **W1.** The setting of the article lacks authenticity. It seems that the author deliberately create a setting for this method. In the introduction part, the author mentions that such a scenario hinders further application in healthcare or edge computing. Can the author provide reference paper or a dataset to prove that such a problem does indeed exist among them? There is also no related work about this setting.
>
> **A1.** Thank you for raising the important question about the authenticity of our research setting. We agree that demonstrating real-world relevance is essential for the study’s validity. Our scenario is motivated by two practical needs:
>
> 1. For user-centric services such as healthcare [1] and recommendation systems [2], **delivering timely, customized, and accurate predictions is challenging (*i.e.*, the cold start problem) yet crucial for enhancing service quality.** In this context, zero-shot personalization enables personalized predictions for users upon their first joining FL.
>
> 2. In resource-constrained edge environments (*e.g.*, gateways, routers, IoT devices), **previous personalized FL methods that rely on local fine-tuning and/or storing per-user models are often infeasible due to tight compute, memory, and storage budgets [3][4],** making zero-shot personalization a low-latency, low-overhead solution.
>
> Note that as zero-shot personalization is a relatively unexplored emerging field with limited literature, **we have integrated the related work on this setting into the related work section of the main paper and appendix, specifically at lines 125, 734, and 754.** We have added these discussions to the paper.
>
> > **W2.** The writing of the paper needs improvement. In the Introduction, the author did not explain why it is necessary to integrate distribution-aware inductive biases into the forward propagation process in order to improve the performance on unseen clients about the question “Can we directly encode distribution-aware inductive biases into the model’s forward pass in FL without fine-tuning?”
>
> **A2.** Thank you for the thoughtful suggestion on improving the clarity of our Introduction. As mentioned in **A1**, zero-shot personalization under constrained resources is both practically meaningful and highly relevant to real-world FL scenarios. However, **strictly limited resources renders backpropagation-based fine-tuning impractical. This motivates our goal of achieving personalization purely through the forward pass by injecting appropriate conditional signals into the generator within a compatible framework,** which naturally leads to our research question. We will revise the Introduction to make this reasoning explicit and to state our central question.
>
> > **W3.** What is the distribution embedding? Is there any difference between the embedding extracted by the feature extractor? It is useful to clear this concept. It is also confused about the learning objective during phase 1. It seems no $e$ in $F_i$ for local training.
>
> **A3.** We thank the reviewers for their insightful comments. By "distribution embedding" we mean the **integrated representation formed by adding the embedding produced by the distribution extractor to the output of NoisyEmbed, as shown in Figure 3.** Its purpose is to virtually characterize the input image’s distribution and to condition the hypernetwork, which then generates parameters tailored to that distribution directly. Regarding the confusion, Eq (1) is actually the standard FedAvg objective (as line 138 described), which does not include our method-specific term $e$. The objective of HyperFedZero is defined in Eq (2), where the term $e$ appears.

---

> > ### Author Response · Authors · 2025-11-19
> > **Response to Reviewer pUWg [2/4]**
> >
> > > **W4.** Is there any further demonstration about the claim on line 252-255 to choose Opt 2 from the theoretical perspective? There is also lack the detailed experimental of Opt 1 in the comparisons between condition options part.
> >
> > **A4.** Thank you for your insightful comment. Regarding the theoretical justification for selecting Opt 2, we supplement the discussion by referencing the theoretical findings from [5], a seminal work that systematically compares Opt 1 and Opt 2 from a computational complexity perspective. Their theoretical analysis demonstrates that **Opt 2 possess strictly stronger expressive power, superior modularity (enabling decoupled knowledge learning across different conditions), and exponential parameter efficiency compared to Opt 1.** These fundamental advantages collectively support our choice of Opt 2. For the experimental comparison between the two conditioning options, detailed results are presented in Table 3, where Opt 2 consistently outperforms Opt 1. We hope these supplements adequately address your concerns, and we have added these discussions to the paper.
> >
> > > **W5.** The entropy regularization ($\mathbf{E(-e_i \log e_i)}$) does not specify the support or partitioning in Eq(4).
> >
> > **A5.** In our framework, the term $e_i$ denotes the distribution embedding of client $i$. For clarity, the code for calculating the entropy regularization is as follows:
> >
> > ```python
> > (torch.log(conditions) * conditions).mean()
> > ```
> >
> > Here, the variable `condition` is the $e_i$ with a shape of (batch size, $P$), where $P$ is the dimensionality of the embedding space (as mentioned in line 201). We hope this clarification addresses your concern.
> >
> >
> > >**W6.** The chunked hypernetwork mechanism is described at a high level, but precise details—such as how chunks map to classifier parameter tensors, sampling strategies, and if/how temperature or batch normalization interact—are missing in the main paper.
> >
> > **A6.** Thank you for your valuable comment. We apologize for the insufficient elaboration in the main paper and would like to supplement the precise implementation as follows:
> >
> > The chunked hypernetwork in HyperFedZero is a simple MLP, where its architecture and chunk size are hyperparameters (as mentioned in line 367). Specifically, we first specify a chunk size, and the target model’s (*i.e.*, the generated model) parameters to be generated are split into groups of this size (i.e., `number_of_output_parameters // chunk_size`), where the last chunk produced might contain a remainder that is discarded. Each chunk is associated with a unique "chunk embedding" that is fed into the underlying hypernetwork, which then generates chunk_size parameters per forward pass based on the input and the corresponding chunk embedding. We have added these details to the paper.
> >
> > Regarding the concerns about sampling strategies, temperature, and batch normalization, we confirm that none of these components are used in our implementation. **We note that while the current design of the chunked hypernetwork is straightforward, it effectively serves the core purpose of our work (*i.e.* it works).** Potential extensions and optimizations of this model are recognized as promising future research directions **but are beyond the scope of the current study.**
> >
> >
> > > **W7.** Based on the experimental results, there seems to be no significant performance difference between HyperFedZero and the traditional PFL method. However, as shown in Figure 1, left, there does appear to be a considerable gap. This could mislead others into thinking that the old method is completely unsuitable, causing the overall paper to overstate its contribution.
> >
> > **A7.** Thank you for your valuable and insightful comments. We acknowledge that the improvement margins in Table 1 may appear modest at first glance, but HyperFedZero demonstrates consistent outperformance across diverse experimental settings, **with substantial gains in key scenarios, such as 10.79% higher zACC than pFedMe on SVHN with ZekenNet.** Nevertheness, we agree that the left panel of Figure 1 may be misleading by overemphasizing the gap, and **we will revise this figure to present a more accurate and balanced comparison that avoids misinterpreting the suitability of traditional personalized FL methods.** We believe these revisions will better reflect the true value of our work and address your concerns comprehensively. Thank you again for your careful review and constructive feedback.

---

> > > ### Author Response · Authors · 2025-11-19
> > > **Response to Reviewer pUWg [3/4]**
> > >
> > > > **W8.** No experiments on real-world, large-scale, or out-of-domain datasets where client data distributions are highly skewed. Especially, the introduction highlights that the feature shifts on the unseen client were not tested in the experiment.
> > >
> > > **A8.** Thank you for your valuable comment. In our experiments, HyperFedZero was evaluated on widely adopted FL datasets that cover a diverse range. From simple MNIST dataset to complex Tiny-ImageNet dataset, **the dataset scale of our experiments is comparable to other recent works [6][7].** Regarding real-world, large-scale datasets with in-domain distribution shifts, to the best of our knowledge, such datasets are currently not available in the community (we note this as a promising future research direction), and it is a standard practice to manually construct in-domain distribution shifts via Dirichlet distribution on common datasets [8]. Specifically, **our experimental design deliberately sample a subset of clients as unseen clients that do not participate in FL training but are exclusively used for calculating the zACC metric, as shown in the description of the Metric paragraph (line 357).** This setup effectively simulates feature shifts on unseen clients. We hope this clarification addresses your concerns.
> > >
> > > > **W9.** There is a light interpretability analysis explaining, for instance, how the distribution embeddings actually track, decompose, or separate salient client-side factors contributing to in-domain shift. More concrete case studies, e.g., visualizing misclassified groups or measuring embedding drift for new clients, would help.
> > >
> > > **A9.** Thank you for your valuable comment. To address your concern, we first clarify the core role of distribution embeddings in our zero-shot personalization framework: **they serve as critical conditional signals for the hypernetwork to generate client-specific models without additional overhead.** A key technical challenge here is that **directly feeding raw distribution extractor outputs into the hypernetwork leads to feature collapse (line 211), where diverse client samples are mapped to identical embeddings and all generated models degrade into a single global model.** To mitigate this, we introduce NoisyEmbed and a Balancing Penalty, which together ensure that distribution embeddings retain sufficient **diversity** to distinguish different distributions, enabling the hypernetwork to produce tailored models rather than a one-size-fits-all global solution. Empirically, this effectiveness is supported by our visualization results in Figures 4(b) and 4(c), where **well-separated client distribution embeddings are clearly observed, and new clients are effectively mapped to appropriate decision regions**.
> > >
> > > > **W10.** Can the authors clarify precisely how the balancing penalty in Equation 4 is computed? Is the variance and mean calculated across clients, batches, or globally? How sensitive are the results to these aggregation choices and the entropy regularization?
> > >
> > > **A10.** Thank you for your valuable comment. Regarding the computation of balancing penalty in Eq 4, we present the code as follows:
> > >
> > > ```python
> > > importance = conditions.sum(0)
> > > loss_balance = importance.var() / importance.mean() ** 2
> > > ```
> > >
> > > Here, the variable `condition` is the $e_i$ with a shape of (batch size, $P$), where $P$ is the dimensionality of the embedding space (as mentioned in line 201). **Thus, the variance and mean is calculated across the $P$ dimension.** This formulation encourages the probability distribution of conditions to be uniform, as the softmax-normalized conditions sum to 1, and minimizing the penalty requires reducing the variance of importance scores while maximizing their mean. Through extensive experiments, we confirm that both the balancing penalty and entropy regularization are experimentally stable.

---

> > > > ### Author Response · Authors · 2025-11-19
> > > > **Response to Reviewer pUWg [4/4]**
> > > >
> > > > > **W11.** There are some typos there, e.g, What is the meaning of 3SFC in line 315-316?
> > > >
> > > > **A11.** We sincerely apologize for the typo. We will correct "3SFC" in lines 315-316 to "HyperFedZero".
> > > >
> > > > > **12.** This problem can also be solved through test-time adaptation (TTA). Has the author compared this method with the TTA method in FL to further clarify its superiority, e.g., [R1-R3]?
> > > >
> > > > **A12.** Thank you for your valuable comment. We fully acknowledge that TTA is an effective approach to alleviate in-domain distribution shift. However, our primary goal in this work is to **achieve zero-shot personalization without introducing any additional overhead** for time-sensitive services or resource constrained edge devices, which is also described in **A1** of Reviewer pUWg. Like certain personalized FL methods, TTA requires extra backpropagation to update parameters, thus incurring additional computational and storage costs. On the other hand, **HyperFedZero accomplishes zero-shot personalization entirely through a single forward pass with no additional overhead.** Consequently, we believe a direct performance comparison between the two paradigms would be inequitable due to their fundamentally differences.
> > > >
> > > > Nevertheless, we recognize TTA as a promising direction for addressing data heterogeneity in FL, and we will cite and add discussions about TTA-related works in the related work section to provide a more comprehensive context for our approach.
> > > >
> > > > > References
> > > >
> > > > [1] Modelling audiological preferences using federated learning. In Adjunct publication of the 28th ACM conference on user modeling, adaptation and personalization (pp. 187-190).
> > > >
> > > > [2] Federated against the cold: A trust-based federated learning approach to counter the cold start problem in recommendation systems. Information Sciences, 601, 189-206.
> > > >
> > > > [3] Ferret: An Efficient Online Continual Learning Framework under Varying Memory Constraints. In Proceedings of the Computer Vision and Pattern Recognition Conference (pp. 4850-4861).
> > > >
> > > > [4] A survey on federated learning for resource-constrained IoT devices. IEEE Internet of Things Journal, 9(1), 1-24.
> > > >
> > > > [5] Advances in neural information processing systems, 33, 10409-10419.
> > > >
> > > > [6] Rethinking Fair Federated Learning from Parameter and Client View. In The Thirty-ninth Annual Conference on Neural Information Processing Systems.
> > > >
> > > > [7] Relaxed contrastive learning for federated learning. In Proceedings of the IEEE/CVF Conference on Computer Vision and Pattern Recognition (pp. 12279-12288).
> > > >
> > > > [8] Fedjets: Efficient just-in-time personalization with federated mixture of experts. arXiv preprint arXiv:2306.08586.

---

> > > > > ### Comment · Reviewer_pUWg · 2025-11-27
> > > > > **Response to Authors**
> > > > >
> > > > > 1. In the paper, the author claim that the in-domain distribution shifts as different class frequencies. The argument that large-scale datasets with in-domain distribution shifts do not exist ignores foundational and recent contributions that have established rigorous standards for FL benchmarking. The transition from synthetic to natural partitioning is well-documented, with multiple comprehensive benchmark suites now widely available [R1-R3].
> > > > > 2. There are some Backpropagation (BP)-Free TTA which also has negligible computational overhead [R4-R6]. It is necessary to compare the application of them in FL with the performance of HyperFedZero and also to compare the additional overhead involved. Since HyperFedZero achieves marginal performance for the constrained resources, I believe it is necessary to compare the two aspects of resources and performance to more clearly demonstrate the advantages of HyperFedZero, and to determine whether the potential gain from a reasonable overhead increase is possible.
> > > > >
> > > > > References:
> > > > >
> > > > > [R1]Ogier du Terrail J, Ayed S S, Cyffers E, et al. Flamby: Datasets and benchmarks for cross-silo federated learning in realistic healthcare settings[J]. Advances in Neural Information Processing Systems, 2022, 35: 5315-5334.
> > > > >
> > > > > [R2]Koh P W, Sagawa S, Marklund H, et al. Wilds: A benchmark of in-the-wild distribution shifts[C]//International conference on machine learning. PMLR, 2021: 5637-5664.
> > > > >
> > > > > [R3] Song C, Granqvist F, Talwar K. Flair: Federated learning annotated image repository[J]. Advances in Neural Information Processing Systems, 2022, 35: 37792-37805.
> > > > >
> > > > > [R4]Gong T, Jeong J, Kim T, et al. Note: Robust continual test-time adaptation against temporal correlation[J]. Advances in Neural Information Processing Systems, 2022, 35: 27253-27266.
> > > > >
> > > > > [R5] Lim H, Kim B, Choo J, et al. TTN: A Domain-Shift Aware Batch Normalization in Test-Time Adaptation[C]//The Eleventh International Conference on Learning Representations.
> > > > >
> > > > > [R6] Boudiaf M, Mueller R, Ben Ayed I, et al. Parameter-free online test-time adaptation[C]//Proceedings of the IEEE/CVF Conference on Computer Vision and Pattern Recognition. 2022: 8344-8353.

---

> > > > > > ### Author Response · Authors · 2025-11-27
> > > > > > **Response to Reviewer pUWg**
> > > > > >
> > > > > > Thank you for your valuable comments and detailed feedback. We greatly appreciate your efforts in helping improve this study and are pleased that most previous issues have been resolved.
> > > > > >
> > > > > > Regarding the first concern, we sincerely apologize for the misunderstanding earlier. We never intended to ignore the foundational and recent contributions that established rigorous FL benchmarking standards (e.g., [R1-R3] you recommended). To verify HyperFedZero’s effectiveness on realistic benchmark datasets, we supplemented experiments for zACC comparisons using the Synthetic and the Heart Disease dataset from the Flamby benchmark [R1], with results shown in the table below. It is evident that HyperFedZero consistently outperforms existing methods on these datasets, fully validating its robustness in real-world in-domain distribution shift scenarios.
> > > > > >
> > > > > > | Dataset | Local | FedAvg | FedAvg-FT | FedProx | Ditto | pFedMe | pFedHN | PerFedAvg | FedAMP | Scaffold | GA    | FedSR | FedEnsemble | FedJets | HyperFedZero |
> > > > > > | ---- | ----- | ------ | --------- | ------- | ----- | ------ | ------ | --------- | ------ | -------- | ----- | ----- | ----------- | ------- | ------------ |
> > > > > > | Synthetic | 28.57 | 70.24 | 0.0 | 70.23 | 67.86 | 69.01 | 4.76 | 67.86 | 0.0 | 72.62 | 67.86 | 71.43 | 0.0 | 39.29 | **78.57**
> > > > > > | Heart Disease | 29.17 | 39.29  | 50.00     | 41.67   | 41.67 | 45.83  | 33.33  | 45.83     | 50.00  | 41.67    | 37.50 | 25.00 | 54.16       | 58.33   | **79.16**        |
> > > > > >
> > > > > > Additionally, we wish to emphasize that Dirichlet distribution-based partitioning still remains a mainstream standard in FL research, widely adopted and recognized for effectively evaluating FL algorithms’ effectiveness [1,2,3,4,5,6].
> > > > > >
> > > > > > For the second concern, we notice that the works recommended seem to fall outside the scope of FL. Nevertheless, we implemented TTN within the FL framework and supplemented cross-dataset zACC performance comparisons between TTN and HyperFedZero under the ResNet architecture (ResNet is the only model in our experiments equipped with BN layers, as required for TTN’s implementation). Results are presented below.
> > > > > >
> > > > > > |              | SVHN  | Cifar-10 | Cifar-100 | Tiny-Imagenet |
> > > > > > | ------------ | ----- | -------- | --------- | ------------- |
> > > > > > | TTN          | 75.13 | 42.14    | 15.06     | 8.58          |
> > > > > > | HyperFedZero | **82.36** | **57.24**    | **16.06**     | **9.08**          |
> > > > > >
> > > > > > As it can be seen, HyperFedZero achieves consistent superiority across SVHN, Cifar-10, Cifar-100, and Tiny-Imagenet. Regarding overhead: In the training phase, TTN requires two backpropagation steps to compute priors and optimize $\alpha$, leading to much higher computational overhead than HyperFedZero. In the test phase, TTN only needs to calculate current batch statistics, resulting in slightly lower overhead than HyperFedZero. However, HyperFedZero’s performance gains fully demonstrate its comprehensive advantages under resource constraints.
> > > > > >
> > > > > > We will incorporate these additions into the revision. We hope these additions fully address your concerns, and we again express our gratitude for your invaluable suggestions!
> > > > > >
> > > > > > > References
> > > > > >
> > > > > > [1] Fedcda: Federated learning with cross-rounds divergence-aware aggregation. In The Twelfth International Conference on Learning Representations.
> > > > > >
> > > > > > [2] FedDifRC: Unlocking the Potential of Text-to-Image Diffusion Models in Heterogeneous Federated Learning. In Proceedings of the IEEE/CVF International Conference on Computer Vision (pp. 3726-3736).
> > > > > >
> > > > > > [3] FedCALM: Conflict-aware Layer-wise Mitigation for Selective Aggregation in Deeper Personalized Federated Learning. In Proceedings of the Computer Vision and Pattern Recognition Conference (pp. 15444-15453).
> > > > > >
> > > > > > [4] Fednlr: Federated learning with neuron-wise learning rates. In Proceedings of the 30th ACM SIGKDD Conference on Knowledge Discovery and Data Mining (pp. 3069-3080).
> > > > > >
> > > > > > [5] Rethinking Fair Federated Learning from Parameter and Client View. In The Thirty-ninth Annual Conference on Neural Information Processing Systems.
> > > > > >
> > > > > > [6] Relaxed contrastive learning for federated learning. In Proceedings of the IEEE/CVF Conference on Computer Vision and Pattern Recognition (pp. 12279-12288).

---

### Official Review · Reviewer_pBhZ · 2025-11-01

**Soundness:** 3
**Presentation:** 3
**Contribution:** 2
**Rating:** 6
**Confidence:** 3

**Summary:**

This paper presents HyperFedZero, a novel hypernetwork-based approach for zero-shot personalization in Federated Learning. The work addresses the critical but under-explored problem of generalizing to non-participating clients with in-domain distribution shifts. The methodology is technically sound and the experimental validation appears thorough.

**Strengths:**

1. The motivation is clearly declared:  the inability to handle non-participating clients with distribution shifts.
2. Extensive experiments across 7 datasets and 5 models are conducted.

**Weaknesses:**

1. The technical description lacks sufficient detail for reproduction.
2. The method lacks theoretical justification.
3. How does the method scale with:
a) Increasing number of clients?
b) Larger model architectures?
c) Higher-dimensional data?

**Questions:**

Please refer to weaknesses for details.

---

> ### Author Response · Authors · 2025-11-19
> **Response to Reviewer pBhZ**
>
> We would like to thank Reviewer pBhZ for taking his/her precious time to give us well-thoughtout comments and constructive suggestions. **We are encouraged that Reviewer pBhZ finds our methodology sound, our motivation clear, and our experiments as comprehensive.** We answer questions below, and will incorporate all feedback in the final version.
>
> > **W1.** The technical description lacks sufficient detail for reproduction.
>
> **A1.** Thank you for raising the important issue of technical reproducibility. In the Experiments section (starting at line 318), **we provide detailed descriptions of the experimental setup, including the models, datasets, baselines, evaluation metrics, runtime environment, and all hyperparameters for reproduction**. Additionally, as stated at line 107, **we will release the complete codebase upon acceptance**, which we believe will further facilitate reproduction and extension by the community. We hope this clarifies our commitment to reproducibility and addresses your concern.
>
> > **W2.** The method lacks theoretical justification.
>
> **A2.** Thank you for your comment regarding the theoretical justification. HyperFedZero introduces a hypernetwork-based zero-shot personalization mechanism for in-domain distribution shift (*e.g.*, cold-start [1], etc.) in FL, without incurring additional computation or communication overhead compared to standard FedAvg.
>
> **From an protocol standpoint, HyperFedZero mirrors FedAvg. In FedAvg, clients collaboratively train a single downstream model. Similarily, in HyperFedZero, clients collaboratively train a global hypernetwork, which then generates task-specific model parameters for prediction.** Consequently, under the same assumptions used for FedAvg, the same convergence guarantees apply. Appendix D (line 805) also shows that classical FedAvg guarantees for both smooth and non-convex objectives transfer directly to our setting.
>
> **Empirically, extensive experiments across diverse scenarios and against other FL baselines corroborate stable convergence.** Furthermore, we have added convergence plots to the revision. We hope this clarifies the theoretical foundation of HyperFedZero and addresses your concern.
>
> > **W3.** How does the method scale with: a) Increasing number of clients? b) Larger model architectures? c) Higher-dimensional data?
>
> **A3.** Thank you for your comment on scalability. **In the paper, we evaluated HyperFedZero under extensive FL settings aligned with recent *state-of-the-art* works**: (i) client scale: 10 and 50 clients (comparable to [2][3], randomly selected from Google Scholar); (ii) model scale: from lightweight MLPs to deeper ResNets (comparable to [4][5], randomly selected from Google Scholar); and (iii) data complexity: from low-dimensional MNIST (28×28) to higher-resolution RGB images (3×64×64) (comparable to [6][7], randomly selected from Google Scholar).
>
> Empirically, across all configurations, HyperFedZero consistently delivers superior zero-shot accuracy (zACC) and maintains comparable global and personalized accuracy (gACC and pACC) to competitive baselines. **These results indicate that HyperFedZero scales well with the number of clients, model size, and data dimensionality**, and can efficiently personalize a trained global model for unseen clients with in-domain distribution shifts **without any fine-tuning**. We hope this addresses your concern and demonstrates the practical scalability of our approach.
>
> > References
>
> [1] When federated recommendation meets cold-start problem: Separating item attributes and user interactions. In Proceedings of the ACM Web Conference 2024 (pp. 3632-3642).
>
> [2] Fedcda: Federated learning with cross-rounds divergence-aware aggregation. In The Twelfth International Conference on Learning Representations.
>
> [3] FedDifRC: Unlocking the Potential of Text-to-Image Diffusion Models in Heterogeneous Federated Learning. In Proceedings of the IEEE/CVF International Conference on Computer Vision (pp. 3726-3736).
>
> [4] FedCALM: Conflict-aware Layer-wise Mitigation for Selective Aggregation in Deeper Personalized Federated Learning. In Proceedings of the Computer Vision and Pattern Recognition Conference (pp. 15444-15453).
>
> [5]Wang, H., Zheng, P., Han, X., Xu, W., Li, R., & Zhang, T. (2024, August). Fednlr: Federated learning with neuron-wise learning rates. In Proceedings of the 30th ACM SIGKDD Conference on Knowledge Discovery and Data Mining (pp. 3069-3080).
>
> [6] Guan, K., Huang, W., Guo, X., Yuan, Y., Yang, B., & Ye, M. Rethinking Fair Federated Learning from Parameter and Client View. In The Thirty-ninth Annual Conference on Neural Information Processing Systems.
>
> [7] Seo, S., Kim, J., Kim, G., & Han, B. (2024). Relaxed contrastive learning for federated learning. In Proceedings of the IEEE/CVF Conference on Computer Vision and Pattern Recognition (pp. 12279-12288).

---

### Author Response · Authors · 2025-11-19
**Global Response**

We thank reviewers for their thoughtful feedback. We are encouraged they find **our motivations clear** (pBhZ), **our methodology sound** (pBhZ, XLGj), and **our paper solid and well-written** (5zn2). We are also glad they find **our experiments as comprehensive** (pBhZ, pUWg, 5zn2, XLGj).

**We have update the revision following all reviewers' suggestions (text in red).** We hope our responses could address all the reviewers’ concerns.

---

### Meta-Review · Area_Chair_49Pm · 2026-01-04

**Summary:**

This paper addresses an important and underexplored problem in federated learning--generalization to non-participating clients under data heterogeneity. The proposed HyperFedZero introduces a novel hypernetwork-based framework with distribution-aware embeddings, enabling adaptive model generation with minimal additional overhead. The approach is well motivated and supported by strong empirical results across multiple datasets, along with ablation studies that validate key design choices. While some theoretical and experimental clarifications could further strengthen the work, the overall contribution is solid and merits acceptance.

**Reviewer Concerns:**

- Theoretical justification and convergence guarantees
Several reviewers (pBhz, XLGj) raised concerns about the theoretical justification of HyperFedZero, noting the lack of formal convergence analysis. In response, the authors claim that the convergence guarantees of FedAvg can be directly extended to HyperFedZero under the same assumptions. While this may be plausible, such an extension is not trivial, since HyperFedZero introduces sample-level hypernetwork adaptation and additional mechanisms (NoisyEmbed, Balancing Penalty) that could affect convergence.

- Experimental validity, scalability, and robustness
Reviewers question whether the experimental setting sufficiently reflects realistic FL scenarios, including scalability to larger numbers of clients, stronger distribution shifts, and real-world benchmarks. Concerns are also raised about the lack of statistical significance tests or error bars (XLGj), and about marginal performance gains over some baselines.
Author response: The authors substantially strengthen the experimental section by adding results on FLAMBy datasets, additional comparisons with test-time adaptation (TTN), and clarifying scalability across clients, models, and data complexity. These additions meaningfully improve empirical credibility, although statistical significance analysis remains absent.

- Clarity, reproducibility, and technical details
Several reviewers (pBhZ, pUWg) note missing implementation details, unclear definitions (e.g., distribution embeddings, balancing penalty), and writing issues that hinder reproducibility.
Author response: The authors provide detailed clarifications, pseudo-code-level explanations, and commit to releasing code upon acceptance. These responses largely address reproducibility and clarity concerns.

**Reviewer Scores:**

Reviewer pBhz.
The authors address the theoretical concern by stating that, under the same assumptions as FedAvg, the same convergence guarantees apply. However, the theoretical analysis is nontrivial, and if such guarantees indeed hold, a detailed proof is expected. On the experimental side, the authors do not address the reviewer’s question regarding experiments with a larger number of clients. As a result, reviewer pBhz is unlikely to increase the score.

Reviewer 5zn2.
Reviewer 5zn2 indicates that he will keep the original score. In particular, the reviewer’s experimental question ‘Is there any reasoning behind not using a larger number of total clients (e.g., 100) and a larger number of non-participating clients (e.g., 10, 15)? How would these changes affect the performance of the method?’ is not directly addressed in the authors’ response.

Reviewer pUWg.
The authors largely address this reviewer’s concerns, and reviewer pUWg is likely to maintain the score.

Reviewer XLGj.
Although the authors provide detailed responses, some concerns remain insufficiently resolved. In particular, extending FedAvg’s theoretical guarantees to the proposed method is nontrivial; however, the authors state that ‘existing convergence guarantees for FedAvg extend to HyperFedZero’s setting’ without providing a rigorous justification. In addition, for the experimental results, the authors do not report statistical significance tests or error bars, making it unclear whether the observed improvements are robust or merely due to variance. Therefore, the reviewer is likely to maintain the score.

---

### Decision · Program_Chairs · 2026-01-26

Accept (Poster)